# TRIM37 prevents formation of centriolar protein assemblies by regulating Centrobin

**Fernando R Balestra[1,2]\*, Andrés Domínguez-Calvo[1,2], Benita Wolf[3†], Coralie Busso[3], Alizée Buff[3], Tessa Averink[3], Marita Lipsanen-Nyman[4], Pablo Huertas[1,2], Rosa M Ríos[2], Pierre Gönczy[3]\***

[1]Departamento de Genética, Universidad de Sevilla, Sevilla, Spain; [2]Centro Andaluz de Biología Molecular y Medicina Regenerativa-CABIMER, Universidad de Sevilla-CSIC-Universidad Pablo de Olavide, Sevilla, Spain; [3]Swiss Institute for Experimental Cancer Research (ISREC), School of Life Sciences, Swiss Federal Institute of Technology Lausanne (EPFL), Lausanne, Switzerland; [4]Pediatric Research Center, Children's Hospital, University of Helsinki and Helsinki University Hospital, Helsinki, Finland

**Abstract** TRIM37 is an E3 ubiquitin ligase mutated in Mulibrey nanism, a disease with impaired organ growth and increased tumor formation. TRIM37 depletion from tissue culture cells results in supernumerary foci bearing the centriolar protein Centrin. Here, we characterize these centriolar protein assemblies (Cenpas) to uncover the mechanism of action of TRIM37. We find that an atypical de novo assembly pathway can generate Cenpas that act as microtubule-organizing centers (MTOCs), including in Mulibrey patient cells. Correlative light electron microscopy reveals that Cenpas are centriole-related or electron-dense structures with stripes. TRIM37 regulates the stability and solubility of Centrobin, which accumulates in elongated entities resembling the striped electron dense structures upon TRIM37 depletion. Furthermore, Cenpas formation upon TRIM37 depletion requires PLK4, as well as two parallel pathways relying respectively on Centrobin and PLK1. Overall, our work uncovers how TRIM37 prevents Cenpas formation, which would otherwise threaten genome integrity.

**\*For correspondence:**
fernando.balestra@cabimer.es
(FRB);
pierre.gonczy@epfl.ch (P)

**Present address:** [†]Department of Oncology, Lausanne University Hospital, and Ludwig Institute for Cancer Research, University of Lausanne, Lausanne, Switzerland

**Competing interests:** The authors declare that no competing interests exist.

## Introduction

Centrioles are small evolutionarily conserved cylindrical organelles characterized by nine triplets of microtubules (MTs) arranged with a striking ninefold radial symmetry (reviewed in *Gönczy and Hatzopoulos, 2019*). In addition to MTs, centrioles contain multiple copies of distinct proteins that contribute to their assembly, structure, and function. Centrioles are essential for the formation of cilia and also recruit pericentriolar material (PCM), including the MT nucleator γ-tubulin ring complex, thus forming the centrosome of animal cells (reviewed in *Bornens, 2012*).

Probably because of such important roles, centriole number is tightly regulated, with most cycling cells having two units at cell cycle onset and four units by the time of mitosis (reviewed in *Sullenberger et al., 2020*). Alterations in centriole number can have adverse consequences on cell physiology and genome integrity. Thus, supernumerary centrioles lead to extra cilia and centrosomes (*Duensing et al., 2007*; *Habedanck et al., 2005*; *Mahjoub and Stearns, 2012*), which can be observed also in several human disease conditions, including certain cancer types (reviewed in *Bettencourt-Dias et al., 2011*; *Chavali et al., 2014*; *Gönczy, 2015*; *Nigg and Holland, 2018*; *Nigg and Raff, 2009*). Despite their importance, the mechanisms that prevent the formation of excess centriolar structures remain incompletely understood.

The two resident centrioles present at the onset of the cell cycle differ in their age: whereas the older, mother, centriole is at least two cell generations old, the younger, daughter, centriole was formed in the previous cell cycle. The mother centriole bears distinctive distal and sub-distal appendages that the daughter centriole acquires only later during the cell cycle (reviewed in *Sullenberger et al., 2020*). In human cells, the proximal region of both mother and daughter centrioles in the G1 phase of the cell cycle is encircled by a torus bearing the interacting proteins CEP57/CEP63/CEP152 (*Brown et al., 2013*; *Cizmecioglu et al., 2010*; *Hatch et al., 2010*; *Lukinavičius et al., 2013*; *Sir et al., 2011*) (reviewed in *Banterle and Gönczy, 2017*). The Polo-like-kinase PLK4 is recruited to this torus, where it focuses to a single location toward the G1/S transition, owing notably to a protective interaction with its substrate STIL, thus marking the site of procentriole assembly (*Klebba et al., 2015*; *Moyer et al., 2015*; *Ohta et al., 2014*) (reviewed in *Arquint and Nigg, 2014*).

The onset of procentriole assembly entails the formation of a ninefold radially symmetric cartwheel thought to act as a scaffold for the entire organelle (reviewed in *Guichard et al., 2018*; *Hirono, 2014*). The fundamental building block of the cartwheel is HsSAS-6, the homologues of which can self-assemble in vitro into ninefold radially symmetric structures akin to those found in vivo (*Guichard et al., 2017*; *Kitagawa et al., 2011b*; *Strnad et al., 2007*; *van Breugel et al., 2011*). During S/G2, the emerging procentriole remains closely associated with the resident centriole and elongates, notably through the contribution of the centriolar proteins CPAP/SAS-4, SPICE, and C2CD3 (*Balestra et al., 2013*; *Comartin et al., 2013*; *Kohlmaier et al., 2009*; *Schmidt et al., 2009*; *Tang et al., 2009*; *Thauvin-Robinet et al., 2014*). During mitosis, the procentriole disengages from the resident centriole in a manner that requires the activity of the Polo-like-kinase PLK1 (*Loncarek et al., 2010*; *Tsou et al., 2009*). Excess PLK1 during S or G2 leads to premature centriole disengagement and centriole reduplication (*Loncarek et al., 2010*; *Tsou et al., 2009*). Normally, disengagement during mitosis generates two centriolar units that are thus licensed to recruit PCM and to trigger a new round of centriole assembly in the following cell cycle.

Centrioles can also assemble independently of a resident centriole. Such de novo assembly occurs in some physiological conditions, for instance when the protist *Naegleria gruberi* transitions from an acentriolar amoeboid life form to a flagellated mode of locomotion (*Fritz-Laylin et al., 2016*; *Fulton and Dingle, 1971*). Likewise, centrioles assemble de novo at the blastocyst stage in rodent embryos (*Courtois et al., 2012*). De novo assembly of centrioles can also be triggered experimentally in human cells following removal of resident centrioles through laser ablation or chronic treatment with the PLK4 inhibitor Centrinone followed by drug release (*Khodjakov et al., 2002*; *Wong et al., 2015*). These findings demonstrate that in human cells de novo assembly is normally silenced by the resident centrioles. Moreover, in contrast to the situation in physiological conditions, experimentally provoked de novo centriole assembly in human cells is error prone and lacks number control (*La Terra et al., 2005*; *Wong et al., 2015*). Furthermore, upon depletion of the intrinsically disordered protein RMB14 or the Neuralized Homology repeat containing protein Neurl4, human cells assemble foci de novo that contain some centriolar proteins and which can function as MTOCs (*Li et al., 2012*; *Shiratsuchi et al., 2015*). Such extra foci, although not bona fide centrioles as judged by electron-microscopy, threaten cell physiology and could conceivably contribute to some disease conditions.

TRIM37 is a RING-B-box-coiled-coil protein with E3 ubiquitin ligase activity (*Kallijärvi et al., 2005*; *Kallijärvi et al., 2002*), which somehow prevents the formation of foci bearing centriolar markers (*Balestra et al., 2013*). Individuals with loss-of-function mutations in both copies of TRIM37 are born with a rare disorder known as Mulibrey nanism (Muscle-liver-brain-eye nanism). The main features of this disorder are growth failure with prenatal onset, as well as characteristic dysmorphic traits and impairment in those organs that give rise to the name of the condition (*Avela et al., 2000*). In addition, Mulibrey patients have a high probability of developing several tumor types (*Karlberg et al., 2009*). Mice lacking Trim37 recapitulate several features of Mulibrey nanism, including a higher propensity to form tumors (*Kettunen et al., 2016*). However, the cellular etiology of Mulibrey nanism remains unclear, partially because of the many roles assigned to this E3 ubiquitin ligase. In tissue culture cells, TRIM37 mono-ubiquitinates and thereby stabilizes PEX5, promoting peroxisomal function (*Wang et al., 2017*). However, Trim37 knock out mice and mouse cell lines depleted of Trim37 do not exhibit peroxisomal associated phenotype (*Wang et al., 2017*),

suggesting that the conserved pathological features exhibited by the mouse disease model must have a different cellular etiology.

The chromosomal region 17q23 where TRIM37 resides is amplified in ~40% of breast cancers (*Sinclair et al., 2003*). TRIM37 mono-ubiquitinates histone H2A in the MCF-7 breast cancer cell line, and this has been reported to dampen the expression of thousands of genes, including tumors suppressors, thus offering a potential link between TRIM37 overexpression and tumorigenesis (*Bhatnagar et al., 2014*). Furthermore, TRIM37 overexpression has been linked to increased cell invasion and metastasis in colorectal and hepatocellular carcinoma (*Hu and Gan, 2017*; *Jiang et al., 2015*). Cancer cells overexpressing TRIM37 are hypersensitive to the absence of centrioles upon treatment with the PLK4 inhibitor Centrinone, because excess TRIM37 interferes with acentriolar spindle assembly, inducing mitotic failure (*Meitinger et al., 2020*; *Yeow et al., 2020*). Moreover, such cells assemble small condensates harboring the centrosomal proteins CEP192 and CEP152, as well as inactive PLK4 (*Meitinger et al., 2020*; *Meitinger et al., 2016*). Interestingly, in addition, the absence of TRIM37 triggers the formation of larger condensates containing PLK4 (*Meitinger et al., 2020*; *Meitinger et al., 2016*). Overall, both depletion and excess of TRIM37 is accompanied by detrimental cellular consequences.

We previously performed a genome-wide siRNA-based screen in human cells to identify regulators of centriole assembly, using the number of foci harboring the centriolar marker Centrin-1:GFP as a readout (*Balestra et al., 2013*). In this screen, we identified TRIM37 as a potent negative regulator of Centrin-1:GFP foci number. Our initial characterization of the TRIM37 depletion phenotype revealed that ~50% of cells possessed supernumerary foci harboring the centriolar proteins Centrin and CP110. Moreover, instances of multipolar spindle assembly and chromosome miss-segregation were observed. Additionally, we found that inhibition of PLK1 partially suppressed supernumerary foci formation upon TRIM37 depletion, leading to the suggestion that such foci occurred through centriole reduplication (*Balestra et al., 2013*), although the fact that suppression was only partial indicated that an additional explanation was to be found. Here, we set out to further explore the nature of such supernumerary foci to uncover the mechanism of action of TRIM37, and thereby perhaps also provide novel insights into Mulibrey nanism.

## Results

### TRIM37 prevents formation of centriolar protein assemblies (Cenpas)

To further decipher the origin of the supernumerary foci containing Centrin and CP110 that appear following TRIM37 depletion, we investigated where in the cell they first occurred. We reasoned that appearance of supernumerary foci close to resident centrioles could be indicative of centriole reduplication, with premature disengagement leading to the licensing of resident centrioles and procentrioles to prematurely seed centriole assembly. By contrast, appearance of supernumerary foci away from resident centrioles could suggest some type of de novo process. We performed live imaging of HeLa cells expressing Centrin-1:GFP (referred to as HC1 cells hereafter) and depleted of TRIM37 by siRNAs. Western blot analyses established that TRIM37 depletion using siRNA was near complete both in HC1 cells and in HeLa cells used hereafter (*Figure 1—figure supplement 1A,B*). As shown in *Figure 1A*, we found that extra Centrin-1:GFP foci can appear in the vicinity of resident centrioles (yellow arrows, 8/13 foci), as well as far from them (orange arrows, 5/13 foci). These results suggest that extra Centrin-1:GFP foci upon TRIM37 depletion may form both through centriole reduplication and some type of de novo process.

To further investigate this question, we analyzed fixed HC1 cells in the S or G2 phase of the cell cycle with antibodies against GFP to monitor Centrin-1:GFP foci, as well as against CEP63 to mark the proximal region of resident centrioles, and HsSAS-6 to mark procentrioles. As expected, we found that control cells harbored four Centrin-1:GFP foci, two of which were CEP63 positive and two of which were HsSAS-6 positive (*Figure 1—figure supplement 1C*). Strikingly, in cells depleted of TRIM37, we found that in addition to the normal four Centrin-1:GFP foci accompanied by two Cep63 foci and two HsSAS-6 foci, ~90% of extra Centrin-1:GFP foci did not harbor CEP63 or HsSAS-6 (*Figure 1B,C*). For comparison, we likewise analyzed cells arrested in G2 following treatment with the CDK1 inhibitor RO3306, which induces PLK1-dependent centriole reduplication (*Loncarek et al., 2010*). In this case, >90% of extra Centrin-1:GFP foci harbored CEP63 and/or HsSAS-6 (*Figure 1B,*

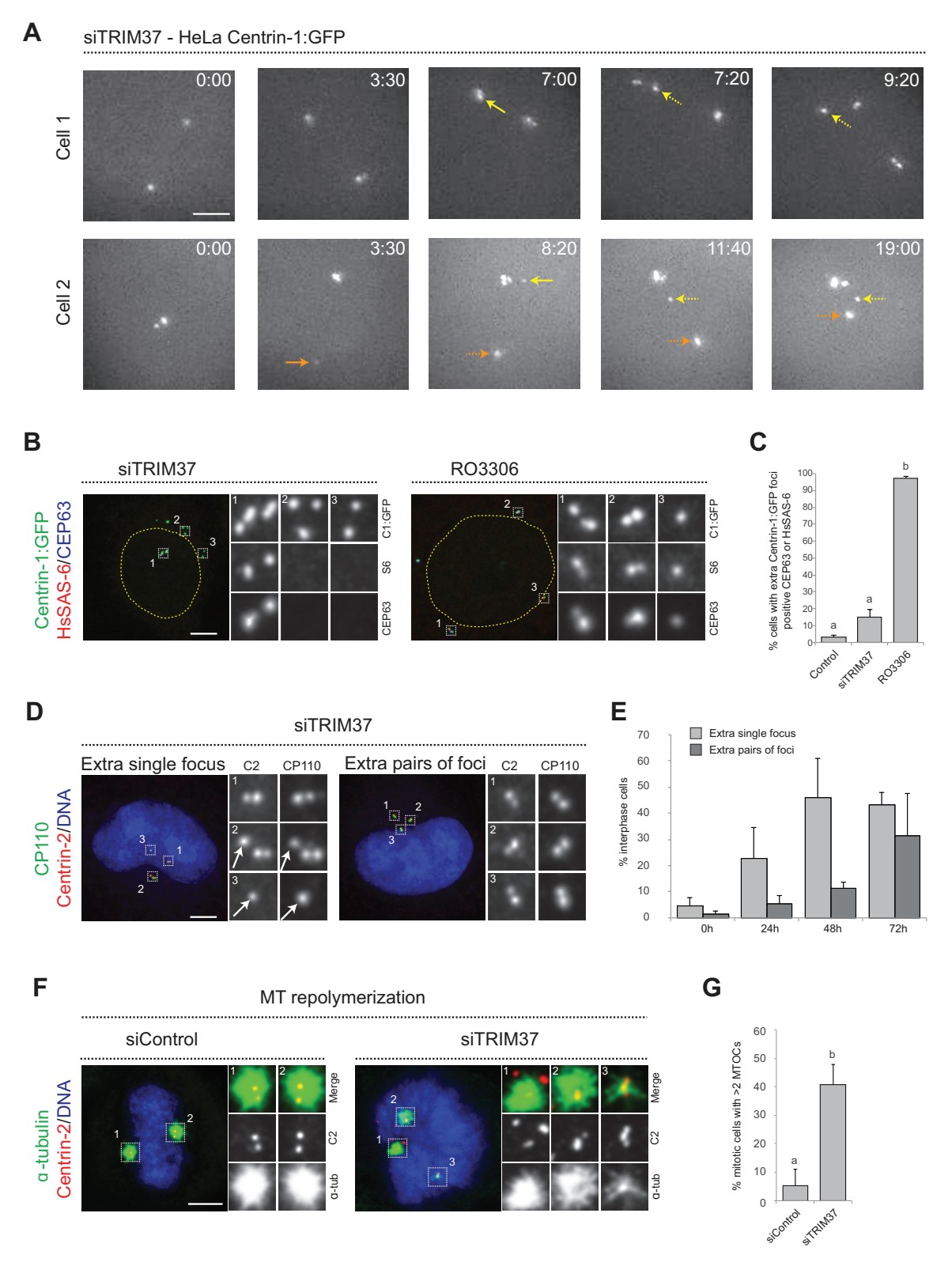

**Figure 1.** Centriolar protein assemblies (Cenpas) form upon TRIM37 depletion. (**A**) Relevant images from wide-field time-lapse recordings of HeLa cells expressing Centrin-1:GFP and depleted of TRIM37 for 48 hr before imaging onset (10 min time frame). Yellow arrows point to two foci appearing close to resident centrioles (8/13 extra foci in 11 cells), orange arrow to one focus appearing away from resident centrioles (5/13 extra foci). Solid arrows indicate first occurrence of foci, dashed arrows their continued presence. Time is indicated in h:min since imaging onset. Note that the intensity of extra

*Figure 1 continued on next page*

*Figure 1 continued*

Centrin-1:GFP foci was typically weaker than that of regular centrioles, especially in the early assembly stages. Note also resident centriole and procentriole appearing in the field of view at the bottom right in Cell 1, 9:20. In this and other Figure panels, scale bars correspond to 5 µm, unless indicated otherwise. (B) HeLa cells expressing Centrin-1:GFP upon treatment with TRIM37 siRNAs or upon RO3306 addition for 48 hr. Cells were immunostained for GFP, HsSAS-6 and CEP63. Nuclear contours are drawn with dashed yellow lines. In this and subsequent figures, magnified images from indicated numbered regions are shown. (C) Corresponding percentage of cells with extra Centrin-1:GFP foci that also harbor CEP63 and/or HsSAS-6. Note that extra Centrin-1:GFP foci could be positive for both Cep63 and HsSAS-6 in RO3306-treated cells. Chart shows the average and SDs from two independent experiments (n = 50 cells each). Here and in other charts of this figure, two conditions that do not share the same letter are significantly different from each other, with p<0.05; unpaired Student's t-test; see *Supplementary file 2* for exact p values. (D) HeLa cells depleted of TRIM37 and immunostained for Centrin-2 plus CP110, illustrating a case with an extra single focus (left, inset 1) and one with an extra pair of foci (right). DNA is shown in blue in this and all other figure panels unless stated otherwise. (E) Corresponding percentage of interphase cells with extra single focus or extra pairs of foci at indicated times after TRIM37 siRNA transfection. Chart shows the average and SDs from three independent experiments (n = 50 cells each). (F) Microtubule depolymerization-regrowth experiment in mitotic HeLa cells treated with control or TRIM37 siRNAs. Microtubules were depolymerized by a 30-min cold shock followed by 1–2 min at room temperature before fixation and immunostaining for Centrin-2 and α-tubulin. (G) Corresponding percentage of mitotic cells with >2 MTOCs. Chart shows the average and SDs from three independent experiments (n = 50 cells each). Note that ~40% of the extra Centrin-2 foci observed in mitosis did not nucleate microtubules, as illustrated for two of them in inset 1 (siTRIM37); data from n = 40 Cenpas in each of the three independent experiments. Source data for panels C, E, and G can be found in *Figure 1—source data 1*. The online version of this article includes the following source data and figure supplement(s) for figure 1:

**Source data 1.** Source data for figure panels: *Figure 1C, E and G*, *Figure 1—figure supplement 2B and E*.
**Figure supplement 1.** TRIM37 depletion and localization of centriolar markers.
**Figure supplement 2.** TRIM37 exerts its centriolar function outside the nucleus and localizes to the distal part of centrioles.

*C*), in contrast to the situation upon TRIM37 depletion. These findings further indicate that TRIM37 does not act solely to prevent centriole reduplication.

We set out to address whether the supernumerary foci that appear following TRIM37 depletion are also active in triggering further rounds of centriole assembly, potentially in a subsequent cell cycle to the one in which they formed. To this end, we transfected cells with TRIM37 siRNAs and analyzed cells 24, 48, and 72 hr thereafter using antibodies against Centrin-2 and CP110. Control cells harbored two individual Centrin-2/CP110 foci in G1 and two pairs of such foci in S/G2, corresponding to two pairs of resident centriole/procentriole (*Figure 1—figure supplement 1D*). Upon TRIM37 depletion, we found that supernumerary Centrin-2/CP110 foci appeared principally as individual units at the 24 hr time point, but that pairs of foci became more frequent thereafter (*Figure 1D,E*). These results indicate that supernumerary Centrin-2/CP110 foci can trigger further rounds of centriole assembly.

Overall, we conclude that TRIM37 depletion results in extra Centrin-1:GFP foci both near and far from resident centrioles, suggestive of centriole reduplication happening together with some de novo process. Moreover, we find that such foci harbor some centriolar proteins but usually not others, and can trigger further rounds of centriole assembly. We will refer hereafter to these entities as *Cen*triolar *p*rotein *as*semblies, or Cenpas in short.

## TRIM37 regulates Cenpas formation from outside the nucleus and localizes to centrosomes

TRIM37 can regulate transcription through nuclear association with the polycomb repressive complex 2 (PRC2) (*Bhatnagar et al., 2014*). To explore whether TRIM37 may function as a transcriptional regulator in preventing Cenpas formation, we addressed whether rescue of the TRIM37 depletion phenotype depended on the presence of the protein in the nucleus. We generated a version of TRIM37 forced to exit the nucleus via fusion to a nuclear export signal (NES). We found that both TRIM37:GFP and TRIM37:NES:GFP equally rescued the TRIM37 depletion phenotype (*Figure 1—figure supplement 2A,B*), indicating that TRIM37 acts outside the nucleus to prevent Cenpas formation.

We explored whether TRIM37 localizes to centrioles. Since antibodies did not prove suitable to address this question (*Balestra et al., 2013*; *Meitinger et al., 2016*), we instead expressed TRIM37:GFP, which was present in the nucleus and more clearly in the cytoplasm (*Figure 1—figure supplement 2C*). Intriguingly, in some cells, TRIM37:GFP also localized to centrosomes marked by γ-tubulin (*Figure 1—figure supplement 2C*). To investigate whether this may reflect a cell cycle restricted

distribution, TRIM37:GFP-expressing cells were probed with antibodies against GFP and Centrobin, which localizes preferentially to the resident daughter centriole and to procentrioles; therefore, G1 cells bear a single Centrobin focus while S/G2 cells bear 2 or 3 (*Zou et al., 2005*; *Figure 1—figure supplement 2D*). This enabled us to establish that whereas only ~10% of G1 cells harbored centrosomal TRIM37:GFP, ~60% of S/G2 cells did so (*Figure 1—figure supplement 2E*). We also localized the fusion protein with respect to CEP63, Centrin-2 and the distal appendage protein CEP164, finding that TRIM37:GFP partially overlapped with CEP164 (*Figure 1—figure supplement 2F,G*). Overall, we conclude that TRIM37 localizes to the distal part of centrioles; it will be interesting to investigate whether TRIM37 acts from this location to prevent the formation of at least some Cenpas, perhaps those near resident centrioles.

## Cenpas can act as MTOCs and are present in Mulibrey patient fibroblasts, resulting in aberrant spindle assembly and chromosome segregation defects

TRIM37-depleted HeLa cells exhibit an increased incidence of multipolar spindles and chromosome miss-segregation (*Balestra et al., 2013*), suggesting that Cenpas can nucleate microtubules and serve as extra MTOCs. To thoroughly test this possibility, we performed microtubule depolymerization-regrowth experiments in HeLa cells depleted of TRIM37 (*Figure 1F*, figure 1 - figure supplement 2H). We found that whereas most control mitotic cells harbored two MTOCs, TRIM37 depletion resulted in an increased frequency of cells with more than two MTOCs, which often differed in size (*Figure 1F,G*). In addition, we found that ~40% of Cenpas did not nucleate microtubules, indicative of some compositional heterogeneity (*Figure 1F*, siTRIM37, inset 1). Overall, we conclude that microtubules nucleated from Cenpas contribute to the aberrant spindle assembly and chromosome miss-segregation phenotypes of TRIM37-depleted cells.

To further explore the importance of Cenpas, we addressed whether they are also present in Mulibrey patient cells. Using healthy donor fibroblasts as controls, we analyzed fibroblasts derived from two patients bearing the Finnish founder mutation, the most frequent TRIM37 disease alteration, which results in a frame shift of the coding sequence that generates a premature stop codon (*Avela et al., 2000*). As reported in *Figure 1—figure supplement 1E*, western blot analysis showed essentially no detectable TRIM37 protein in patient cells. Moreover, we immunostained control and patients fibroblast with antibodies against Centrin-2 to monitor the presence of Cenpas, as well as against γ-tubulin to probe their ability to recruit PCM. Echoing the results in tissue culture cell lines depleted of TRIM37, we found that patient cells in mitosis harbored supernumerary Centrin-2 foci, some of which were positive for γ-tubulin (*Figure 2A,B*). We then analyzed microtubule distribution to assess the impact of Cenpas on mitotic spindle assembly. During metaphase, the spindle was invariably bipolar in control fibroblasts, whereas patient fibroblasts with Cenpas frequently harbored pseudo-bipolar (~25% of cases) and multipolar (~31% of cases) spindles (*Figure 2C–E*). These aberrant figures were corrected in most cases by anaphase, when mitotic spindles were predominantly bipolar (~87% of cases) (*Figure 2F,G*). Importantly, however, even such spindles were not fully functional: chromosome segregation defects revealed by chromosome bridges or lagging chromosomes were observed in ~16% of patient cells with a bipolar anaphase spindle (*Figure 2F,H*). Moreover, micronuclei were present in ~9% of interphase patient fibroblast (*Figure 2I,J*).

Overall, we conclude that Cenpas are present and active in Mulibrey patient cells, thus contributing to genome instability.

## Ultra expansion microscopy and electron microscopy reveal aberrant centriole-related structures upon TRIM37 depletion

We set out to address whether Cenpas exhibit further hallmarks of centrioles. We stained cells depleted of TRIM37 with antibodies against acetylated tubulin, a signature modification of centriolar microtubules, finding that ~23% cells possessed extra acetylated tubulin foci (*Figure 3—figure supplement 1A,B*). To examine this feature at higher resolution, we turned to ultrastructure expansion microscopy (U-ExM) coupled to confocal imaging (*Gambarotto et al., 2019*). Control and TRIM37-depleted RPE-1 cells expressing Centrin1:GFP were immunostained for GFP to identify Cenpas, for CEP152 to mark mature centrioles and for acetylated tubulin. Control cells contained two mature centrioles positive for all three markers (*Figure 3A*). We found that some of the Cenpas formed

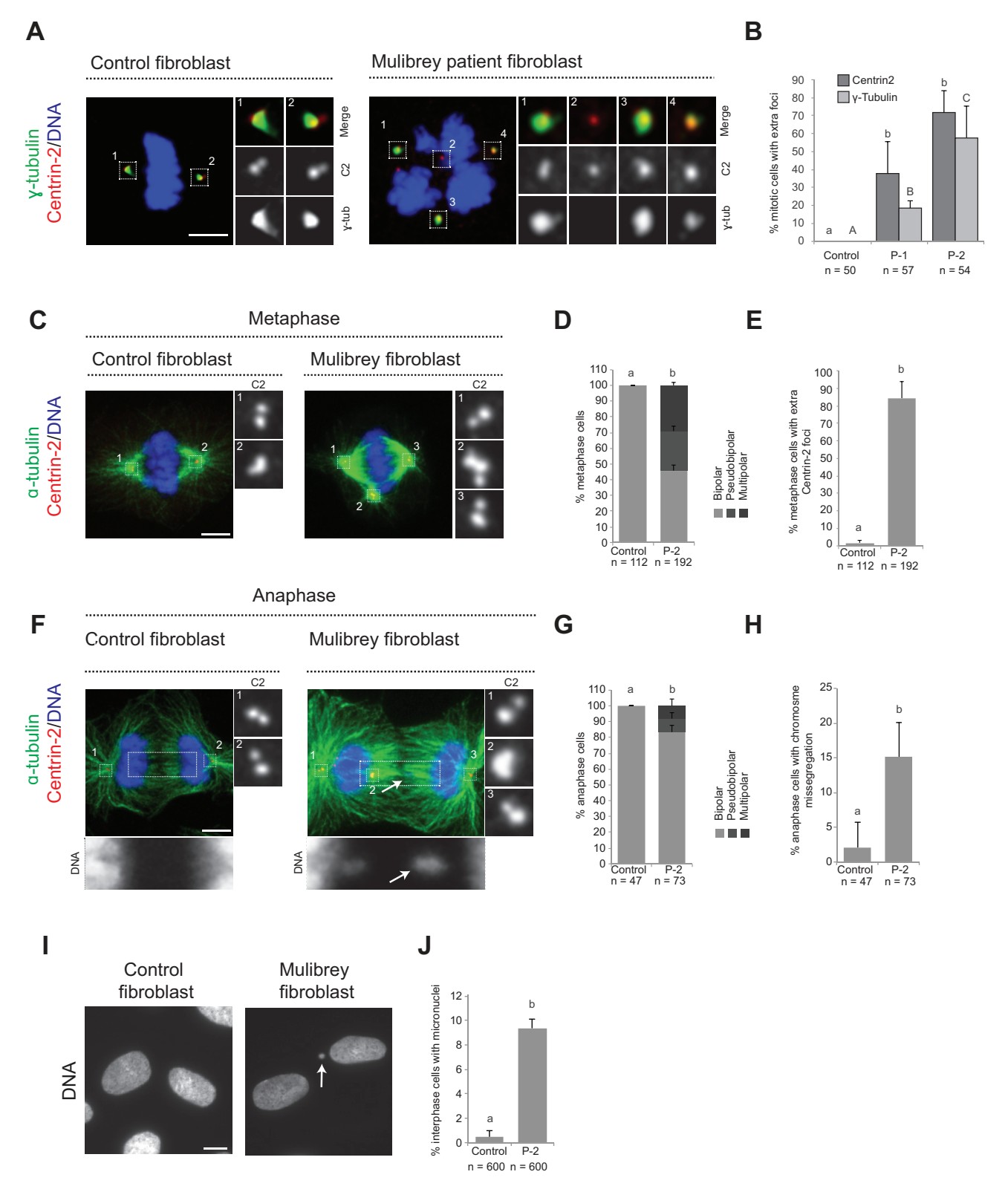

**Figure 2.** Cenpas can behave as extra MTOCs, including in Mulibery patient cells. (**A**) Control and patient-1 (P-1) fibroblasts in mitosis immunostained for Centrin-2 and γ-tubulin. (**B**) Corresponding percentage of mitotic cells with extra number of Centrin-2 or γ-tubulin foci in control and patient (P-1 and P-2) fibroblasts. Chart shows the average and SDs from three independent experiments (n: total number of cells scored per condition). Here and in other charts of this figure, two conditions that do not share the same letter are significantly different from each other, with p<0.05; unpaired Student's

*Figure 2 continued on next page*

*Figure 2 continued*

t-test; see *Supplementary file 2* for exact p values. (**C**) Control and patient-2 (P-2) fibroblasts in metaphase immunostained for Centrin-2 and α-tubulin. (**D, E**) Corresponding percentage of metaphase cells with bipolar, pseudobipolar or multipolar spindles (**D**), and percentage of metaphase cells with extra number of Centrin-2 foci (**E**). Charts show the average and SDs from three independent experiments (n: total number of cells scored per condition). (**F**) Control and patient-2 (P-2) fibroblasts in anaphase immunostained for Centrin-2 and α-tubulin. (**G, H**) Corresponding percentage of anaphase cells with bipolar, pseudobipolar or multipolar spindles (**G**), and percentage of bipolar or pseudobipolar anaphase cells with chromosome segregation defects (**H**). Charts show the average and SDs from three independent experiments (n: total number of cells scored per condition). (**I**) Control and patient-2 (P-2) interphase fibroblasts stained with DAPI. (**J**) Corresponding percentage of interphase cells bearing a micronucleus. Chart shows the average and SDs from three independent experiments (n: total number of cells scored per condition). Source data for panels B, D, E, G, H, and J can be found in *Figure 2—source data 1*.

The online version of this article includes the following source data for figure 2:

**Source data 1.** Source data for figure panels: *Figure 2B, D, E, G, H and J*.

upon TRIM37 depletion harbored merely Centrin1:GFP, but neither acetylated tubulin nor CEP152 (*Figure 3B–D*, yellow arrows). By contrast, other Cenpas were positive for all three markers (*Figure 3C–E*), with the acetylated tubulin signal being sometimes smaller than normal (*Figure 3C, D*, white arrows). Moreover, some Cenpas appeared to have matured into entities with seemingly regular acetylated tubulin and CEP152 signals (*Figure 3E*). Together, these findings support the notion that Cenpas are heterogeneous in nature, with partially overlapping composition.

To uncover the ultrastructure of Cenpas, we conducted correlative light and electron microscopy (CLEM). Using fluorescence microscopy, we screened HeLa and RPE-1 cells expressing Centrin-1:GFP and depleted of TRIM37 to identify Cenpas, using a gridded coverslip to acquire information regarding GFP foci position, and then conducted serial section transmission electron microscopy (TEM). In addition to control cells (*Figure 3—figure supplement 1C,D*), we analyzed eight cells depleted of TRIM37 (*Figure 3F*, *Figure 3—figure supplement 1E,K,R*, *Supplementary file 1*). We observed a total of 47 Centrin-1:GFP foci by light microscopy in these eight TRIM37-depleted cells and found most of the expected resident centrioles (15/16, *Figure 3G,H*; *Figure 3—figure supplement 1F,G,M,Q,S,T*). In addition, this analysis uncovered 22 unusual structures (*Supplementary file 1*). Ten of these were variable centriole-related electron-dense assemblies that harbored microtubules, but only partially resembled centrioles (*Figure 3I,K,L*; *Figure 3—figure supplement 1I,J,L,N, O,P*). Strikingly, the remaining 12 unusual structures were elongated electron-dense striped entities, hereafter referred as 'tiger structures' (*Figure 3J,M*; *Figure 3—figure supplement 1H,U*). We noted also that an individual tiger structure sometimes correlated with more than one Centrin-1:GFP focus (*Figure 3—figure supplement 1U*).

Overall, we conclude that the ultrastructure of Cenpas formed upon TRIM37 depletion is somewhat heterogeneous, perhaps reflecting different pathways or steps in their assembly.

## TRIM37 depletion triggers formation of elongated Centrobin assemblies that likely correspond to the tiger structures

Because TRIM37 is an E3 ligase, the activity of which is important for preventing Cenpas formation (*Balestra et al., 2013*), we reasoned that one or several proteins implicated in centriole assembly might accumulate in an aberrant manner upon TRIM37 depletion, causing the observed phenotype. Therefore, we conducted a small screen by immunostaining cells depleted of TRIM37 with antibodies against >20 centriolar and centrosomal proteins (*Figure 4—figure supplement 1A*). Strikingly, this analysis revealed notably that Centrobin, which normally localizes tightly to the daughter centriole and to procentrioles (*Zou et al., 2005*), is present in elongated assemblies in the cytoplasm upon TRIM37 depletion (*Figure 4A,B*). We found that ~80% of TRIM37-depleted cells usually bore one or two such Centrobin assemblies (*Figure 4C*; *Figure 4—figure supplement 1B*), and that all cells harboring Cenpas had Centrobin assemblies (n = 150), with Cenpas often colocalizing with them (~79%). Moreover, while most centriolar proteins tested did not localize to Centrobin assemblies (*Figure 4—figure supplement 1A and C–E*), we found that SPICE did, although it was not needed for the formation of Centrobin assemblies (*Figure 4—figure supplement 1F,G*). Furthermore, we uncovered that PLK4 is also detected in Centrobin assemblies, which therefore likely coincide with the large PLK4 condensates that form in TRIM37-knock out RPE-1 cells (TRIM37-ko) (*Meitinger et al., 2020*; *Meitinger et al., 2016*; *Figure 4—figure supplement 1H*; Materials and

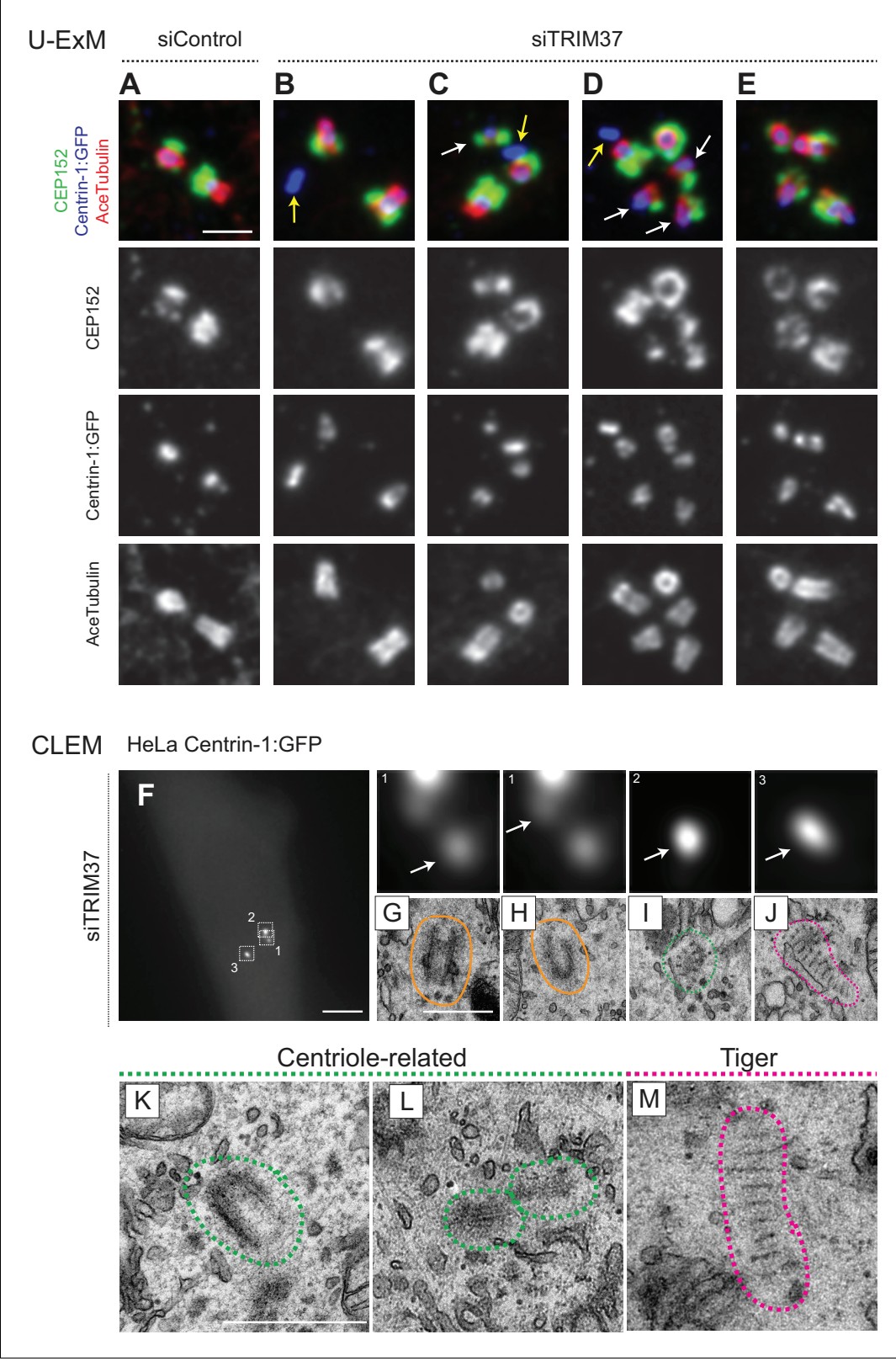

**Figure 3.** Cenpas are structures related to centrioles or electron-dense striped structures. (A–E) Ultrastructure expansion microscopy (U-ExM) confocal images of control (A) or TRIM37 (B–E) depleted RPE-1 cells expressing Centrin-1:GFP, and immunostained for GFP, CEP152 as well as acetylated tubulin. Yellow arrows point to Cenpas lacking CEP152 and acetylated tubulin, white arrows to those harboring both proteins, but with an unusual

*Figure 3 continued on next page*

*Figure 3 continued*

distribution. Scale bar 500 nm. (**F–J**) CLEM analysis of HeLa cell (cell three in *Supplementary file 1*) expressing Centrin-1:GFP and depleted of TRIM37. Maximal intensity projection of wide-field microcopy image covering the entire cell volume (**F**), and magnified insets from the light microscopy images above the corresponding 50 nm section EM images (**G–J**), with white arrows pointing to relevant Centrin-1:GFP focus. Scale bars: 5 µm in F, 500 nm in G. Here and in panels K-M, green and pink dashed lines surround centriole-related and tiger structures, respectively. Filled orange lines surround resident centrioles, which could be recognized when going through all the sections encompassing the organelle. (**K–M**) Centriole-related (K, cell seven in *Supplementary file 1*; L, cell seven in *Supplementary file 1*), and tiger structure (M, cell two in *Supplementary file 1*). Scale bar is 500 nm.

The online version of this article includes the following source data and figure supplement(s) for figure 3:

**Figure supplement 1.** Cenpas formed upon TRIM37 depletion are centriole related structures or striped structures.

**Figure supplement 1—source data 1.** Source data for figure panels: *Figure 3—figure supplement 1B*.

---

methods). Just like for SPICE, we found that Centrobin assemblies generated upon TRIM37 depletion formed even when PLK4 was targeted using siRNAs (*Figure 4D,E*), despite the presence of monopolar mitotic figures attesting to the efficiency of depletion. However, the size of Centrobin assemblies was slightly diminished upon PLK4 depletion (*Figure 4D*; *Figure 4—figure supplement 1I,J*; see Discussion). By contrast, depletion of Centrobin by siRNAs precluded formation of large PLK4 condensates (*Figure 4D,E*).

We set out to further characterize the elongated Centrobin assemblies formed upon TRIM37 depletion. We used U-ExM coupled to STED super-resolution microscopy to analyze these assemblies at higher resolution. We immunostained RPE-1 cells expressing Centrin-1:GFP with antibodies against GFP, CEP152, and Centrobin. In control conditions, centrioles viewed in cross-section exhibited a clear localization of Centrobin between the outer CEP152 and the inner Centrin-1:GFP signals (*Figure 4F*). Cells depleted of TRIM37 exhibited analogous distributions at resident centrioles (*Figure 4F*), but also harbored elongated Centrobin assemblies abutting Centrin-1:GFP foci (*Figure 4F*, arrows). Strikingly, the superior resolution afforded by U-ExM coupled to STED revealed that such Centrobin assemblies were striated (*Figure 4F*). Suggestively, the inter-stripe distances of these Centrobin assemblies were analogous to those of the tiger structures unveiled through CLEM (*Figure 4G*). In summary, U-ExM analysis strongly suggests that Centrobin is a constituent of the electron-dense tiger structures observed by TEM upon TRIM37 depletion, and raises the possibility that such structures serve as platforms for Cenpas formation.

## TRIM37 depletion alters Centrobin stability

How could TRIM37 regulate Centrobin? Performing real time quantitative PCR experiments showed a slight diminution in Centrobin mRNA levels upon TRIM37 depletion (*Figure 4—figure supplement 2A*), suggesting that regulation is not at the transcriptional level. By contrast, western blot analysis uncovered that Centrobin protein levels were slightly increased upon TRIM37 depletion (*Figure 4H*). Given the elongated Centrobin assemblies identified by immunostaining, we speculated that the overall increase in Centrobin protein level might reflect an accumulation into such structures, potentially in an insoluble form. Accordingly, fractionating cell lysates into soluble and insoluble fractions, we found that the increase in Centrobin protein levels was most pronounced in the latter (*Figure 4I*). We noted also that the insoluble pool of Centrobin appeared to migrate slower in the gel upon TRIM37 depletion, suggesting that TRIM37 not only restricts Centrobin levels, but also might somehow regulate its posttranslational state.

Since TRIM37 is an E3 ubiquitin ligase, we reasoned that its activity could modulate Centrobin protein degradation and, thereby, stability. To assess Centrobin protein stability, we monitored Centrobin protein levels by Western blot analysis in the presence of the translation inhibitor Cycloheximide, both in control and TRIM37-depleted cells. As reported in *Figure 4J and K*, we found that TRIM37 depletion increased Centrobin protein stability. One possibility would be that, normally, TRIM37 ubiquitinates Centrobin, thus targeting it for degradation, such that increased Centrobin levels upon TRIM37 depletion would trigger formation of Centrobin assemblies and Cenpas. However, although Centrobin overexpression generates aggregates (*Jeong et al., 2007*), we found that

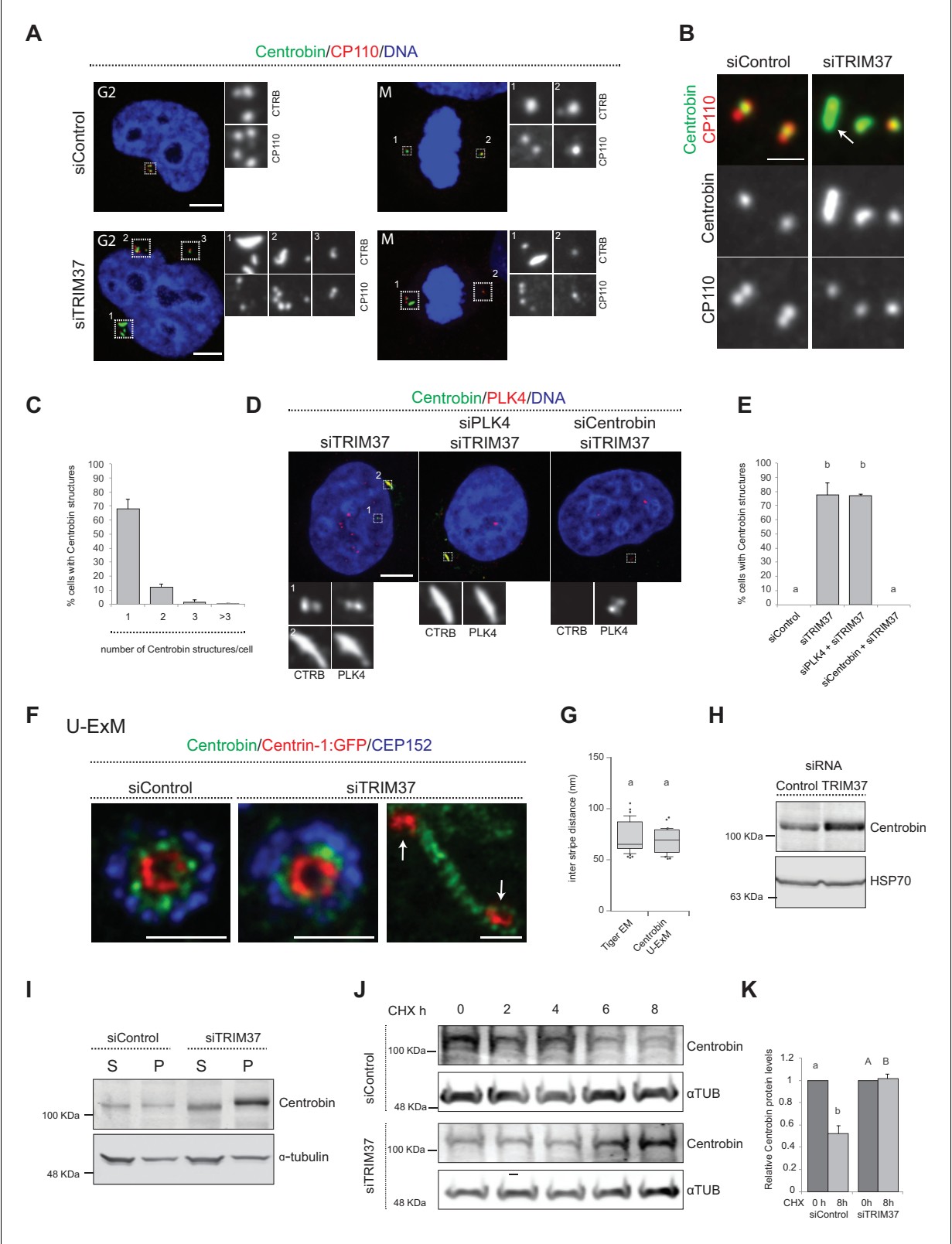

**Figure 4.** TRIM37 regulate Centrobin protein stability and levels. (**A**) HeLa cells in G2 or mitosis (M), as indicated in the upper left corners of the images, treated with control or TRIM37 siRNAs, and immunostained for CP110 plus Centrobin. (**B**) High-magnification confocal view of cells treated with control or TRIM37 siRNAs immunostained for Centrobin and CP110. Arrow points to elongated Centrobin assembly. Scale bar 1 µm. (**C**) Number of Centrobin structures in HeLa cells depleted of TRIM37. Chart shows the average and SDs from three independent experiments (n = 50 cells each). (**D**)

*Figure 4 continued on next page*

*Figure 4 continued*

HeLa cells were first transfected with PLK4 or Centrobin siRNAs and 24 hr thereafter transfected again with TRIM37 siRNAs. Cells were fixed 72 hr after first transfection and stained with Centrobin and PLK4 antibodies (*Wong et al., 2015*). (E) Corresponding percentage of cells bearing Centrobin structures. Chart shows the average and SDs from three independent experiments (n = 50 cells each). Here and in other charts of this figure, two conditions that do not share the same letter are significantly different from each other, with p<0.05; unpaired Student's t-test; see *Supplementary file 2* for exact p values. (F) U-ExM coupled to STED super-resolution microscopy of RPE-1 cells immmunostained for CEP152, Centrin-1 and Centrobin. White arrows point to Cenpas in close proximity to Centrobin assembly. Scale bars are 250 nm. (G) Box-and-whisker plot of inter stripe distances in TEM tiger structures (n = 53 from five tiger structures) and U-ExM Centrobin structures (n = 30 from three Centrobin structures). U-ExM-induced sample expansion was taken into consideration to compare TEM vs. U-ExM inter stripe measurements. Box plots show median, interquartile range (10–90 percentile) and SDs. (H) Western blot of lysates from HeLa cells treated with control or TRIM37 siRNAs probed with antibodies against Centrobin (top) or HSP70 as loading control (bottom). (I) Western blot of soluble (S) or insoluble (P, for pellet) fractions of lysates from HeLa cells treated with control or TRIM37 siRNAs, probed with antibodies against Centrobin (top) or α-tubulin as loading control (bottom). Note that Centrobin in the insoluble fraction migrates slower upon TRIM37 depletion, suggestive of some posttranslational modification. (J) Western blot of total Centrobin protein levels in control and TRIM37-depleted HeLa cells treated with cycloheximide (CHX) for indicated time in hours (h), probed with antibodies against Centrobin (top) or α-tubulin as loading control (bottom). Note that the amount of lysate loaded for the TRIM37-depleted sample was ~50% of that loaded for the siControl condition in this case. (K) Quantification of relative Centrobin protein levels from western blots such as the one shown in J. Chart shows the average and SDs from two independent experiments. Lower-case and upper-case letters above the charts reflect comparisons of two distinct data sets. Source data for panels C, E, G, and K can be found in *Figure 4—source data 1*.

The online version of this article includes the following source data and figure supplement(s) for figure 4:

**Source data 1.** Source data for figure panels: *Figure 4C, E, G and K*, *Figure 4—figure supplement 1B* and *Figure 4—figure supplement 2A*.

**Figure supplement 1.** Distribution of centriolar and centrosomal proteins upon TRIM37 depletion, as well as testing of SPICE and PLK4 requirements for Centrobin assembly formation.

**Figure supplement 2.** Increased Centrobin level is not sufficient for Cenpas formation.

such aggregates did not resemble the Centrobin assemblies uncovered here, nor did they trigger Cenpas formation (*Figure 4—figure supplement 2B*). In addition, TRIM37 overexpression did not alter Centrobin centrosomal distribution (*Figure 4—figure supplement 2C*). Moreover, no evidence for TRIM37-mediated Centrobin ubiquitination was found in cell free assays (data not show), such that the detailed mechanisms of Centrobin modulation by TRIM37 remain to be deciphered. Regardless, we conclude that TRIM37 normally regulates Centrobin stability, preventing the protein from forming the elongated assemblies that are invariably present in cells with Cenpas.

## Centrobin assemblies are present in Mulibrey patient fibroblasts and may serve as platforms for Cenpas formation

We set out to investigate whether Centrobin assemblies are also present in fibroblasts derived from a Mulibrey patient. Importantly, we found that this was the case in the majority of cells (*Figure 5A, B*), with most Centrobin assemblies being associated with Cenpas (~88%, N = 150). Furthermore, we found that Centrobin assemblies are present and usually coincident with active MTOCs during mitosis in patient fibroblasts (*Figure 5C,D*).

To address whether Centrobin might be required for Cenpas formation, we investigated their assembly kinetics following release from a double thymidine block in cells depleted of TRIM37. Initially most cells with elongated Centrobin assemblies lacked Cenpas, whereas at later time point cells with both Centrobin assemblies and Cenpas became more prevalent (*Figure 5E,F*). Therefore, the appearance of elongated Centrobin assemblies precedes that of Cenpas, compatible with the notion that the former is needed for the latter.

To investigate the potential role of Centrobin in Cenpas formation, we tested whether Centrobin depletion reduces Cenpas numbers in cells depleted of TRIM37. Although Centrobin depletion was reported initially to impair centriole assembly in HeLa cells (*Zou et al., 2005*), more recent work with Centrobin knock out cells (Centrobin-ko) demonstrated that the protein is dispensable for this process in RPE-1 cells (*Ogungbenro et al., 2018*). In our hands, siRNA-mediated depletion of Centrobin did not impact centriole assembly in HeLa Kyoto cells either, despite near-complete protein depletion (*Figure 5—figure supplement 1A,B*). As anticipated, Centrobin assemblies disappeared entirely from cells doubly depleted of Centrobin and TRIM37 (*Figure 5—figure supplement 1C*). Importantly, we found that Cenpas number was significantly lowered in such doubly depleted cells compared to cells depleted of TRIM37 alone (*Figure 5—figure supplement 1D*). Interestingly, however, even if Centrobin depletion was complete as judged by western blot analysis (*Figure 5—figure*

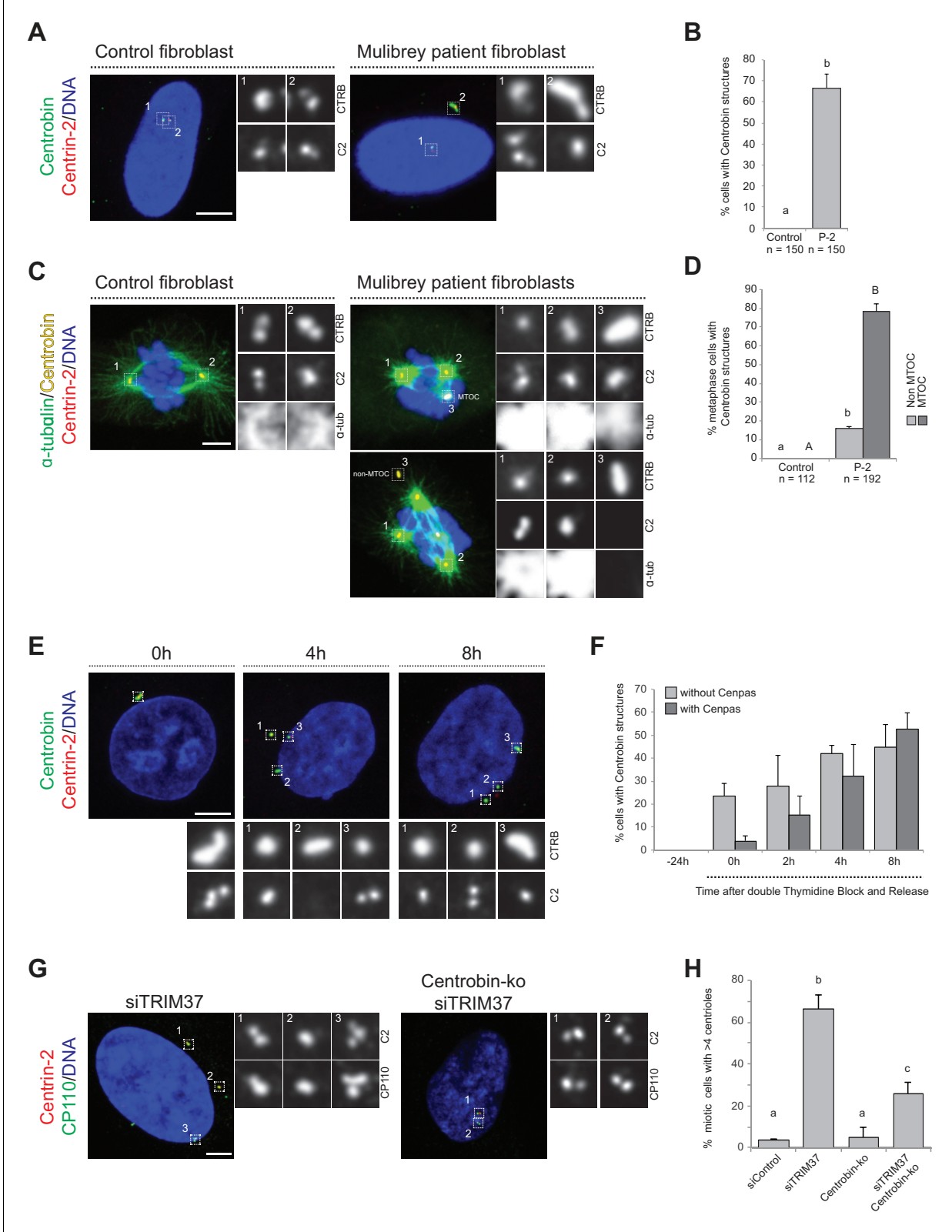

**Figure 5.** Centrobin promotes Cenpas formation. (**A**) Control and patient-2 fibroblasts immunostained for Centrin-2 and Centrobin. (**B**) Corresponding percentage of cells bearing Centrobin structures. Chart shows the average and SDs from three independent experiments (n: total number of cells scored per condition). Here and in other charts of this figure, two conditions that do not share the same letter are significantly different from each other, with p<0.05; unpaired Student's t-test; see *Supplementary file 2* for exact p values. (**C**) Control and patient-2 fibroblasts in mitosis

*Figure 5 continued*

immunostained for Centrin-2, Centrobin and α-tubulin. Note that Centrobin structures can either act as MTOCs (inset 3, top cell) or not (inset 3, bottom cell). (D) Corresponding percentage of metaphase cells with Centrobin structures either associated or not associated to an active MTOC. Chart shows the average and SDs from three independent experiments (n: total number of cells scored per condition). Lower-case and upper-case letters above the charts reflect comparisons of two distinct data sets. (E) HeLa cells were synchronized with a double thymidine block, released and transfected with TRIM37 siRNAs 24 hr before second thymidine release. Cells were fixed and immunostained with antibodies against Centrin-2 and Centrobin at the time of transfection (−24 hr) and at the indicated times after release. (F) Corresponding percentage of cells with Centrobin assemblies either in close proximity to Cenpas or else not associated with them. Chart shows the average and SDs from three independent experiments (n = 50 cells each). (G) Control and Centrobin-ko RPE-1 cells transfected with TRIM37 siRNAs immunostained for Centrin-2 and CP110. (H) Corresponding percentages of mitotic cells with >4 CP110 foci. Chart shows the average and SDs from three independent experiments (n = 50 cells each). Source data for panels B, D, F, and H can be found in *Figure 5—source data 1*.

The online version of this article includes the following source data and figure supplement(s) for figure 5:

**Source data 1.** Source data for figure panels: *Figure 5B, D, F and H*, *Figure 5—figure supplement 1B and D*.
**Figure supplement 1.** Centrobin promotes Cenpas assembly.

*supplement 1E,F*), Cenpas formation upon TRIM37 depletion was only partially prevented by Centrobin siRNA treatment (*Figure 5—figure supplement 1D*). To test whether this might reflect residual Centrobin in the double siRNA depletion setting, we performed a similar experiment with RPE-1 Centrobin-ko cells (*Ogungbenro et al., 2018*), reaching analogous conclusions (*Figure 5G,H*, *Figure 5—figure supplement 1G*). Together, these results support the view that upon TRIM37 depletion Centrobin assemblies act as a platform seeding the formation of some, but not all, Cenpas.

## Centrobin and PLK1 together promote Cenpas assembly upon TRIM37 depletion

To further understand the mechanisms of Cenpas formation upon TRIM37 depletion, we tested if select components that are critical for canonical centriole duplication were also needed for Cenpas generation. To test the role of PLK4 kinase activity, HeLa cells were grown in the presence of Centrinone for 5 days and then depleted of TRIM37 for 3 days in the continued presence of Centrinone. We found that Cenpas did not form under these conditions, demonstrating an essential role for PLK4 kinase activity (*Figure 6A and B*). We also tested the requirement for HsSAS-6, STIL, CPAP, and SPICE. As anticipated, single depletion of these components resulted in decreased centriole number (*Figure 6A*). However, depletion of STIL, CPAP, or SPICE did not dramatically modify the number of Cenpas upon TRIM37 depletion (*Figure 6A*). By contrast, HsSAS-6 depletion reduced Cenpas number, albeit less so than upon Plk4 inactivation (*Figure 6A*). To further explore the impact of HsSAS-6, we depleted TRIM37 from RPE-1 p53-/- HsSAS-6-knock out cells (HsSAS-6-ko) (*Wang et al., 2015*). Although HsSAS-6-ko cells invariably lacked centrioles (*Figure 6A,B*), some Cenpas nevertheless formed upon TRIM37 depletion, although to a lesser extent than following depletion of TRIM37 alone (*Figure 6A,B*). In agreement with the absence of HsSAS-6 in elongated Centrobin assemblies (see *Figure 4—figure supplement 1E*) and the fact that PLK4 depletion does not impact such assemblies (see *Figure 4D*), we found that elongated Centrobin assemblies were generated unabated upon TRIM37 depletion in cells treated with Centrinone or lacking HsSAS-6 (*Figure 6C,D*; *Figure 6—figure supplement 1A*). We conclude that PLK4 activity and HsSAS-6 act downstream of, or in parallel to, Centrobin in the pathways leading to Cenpas formation upon TRIM37 depletion.

To further understand the requirements for Cenpas generation, considering that PLK1 had been shown to contribute partially to their formation (*Balestra et al., 2013*), and that we found here the same to be true for Centrobin, we set out to investigate whether the combined removal of PLK1 and Centrobin fully prevents Cenpas generation. To avoid the negative impact of PLK1 inhibition on cell cycle progression, we performed these experiments in synchronized cells depleted of TRIM37, and monitored Cenpas appearance during G2 after release from an S phase arrest. Cells were also subjected to Centrobin depletion and/or BI-2536 treatment to inhibit PLK1. We ascertained by FACS analysis that cell cycle progression was not blocked in these conditions (*Figure 6—figure supplement 1B*). Importantly, we found that simultaneous Centrobin depletion and PLK1 inhibition completely prevented Cenpas formation (*Figure 6E*), indicating that PLK1 and Centrobin act in parallel to promote Cenpas formation upon TRIM37 depletion.

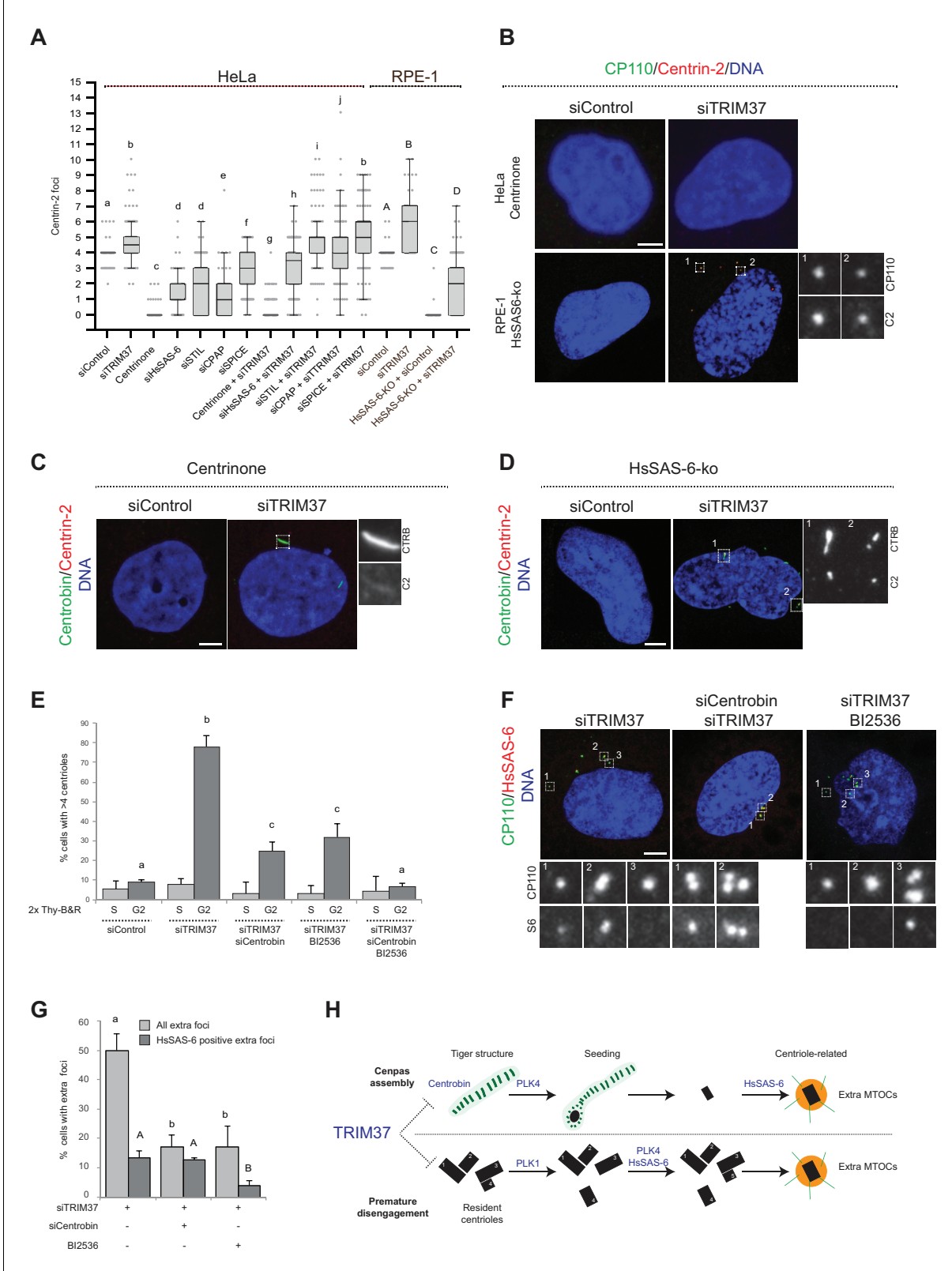

**Figure 6.** Two pathways contribute to Cenpas formation upon TRIM37 depletion. (**A**) Box-and-whisker Tukey plot of Centrin-2 foci number per cell in indicated conditions. Box plots show median, interquartile range (10–90 percentile) and SDs from two independent experiments (n = 50 cells each). All cells were analyzed in mitosis with the exception of HsSAS-6-ko conditions. Here and in other charts of this figure, two conditions that do not share the same letter are significantly different from each other, with p<0.05; unpaired Student's t-test; see ***Supplementary file 2*** for exact p values. Lower-case

*Figure 6 continued on next page*

*Figure 6 continued*

and upper-case letters above the charts reflect comparisons of two distinct data sets. (B) HeLa cells grown with centrinone for 8 days (top) or RPE-1 HsSAS-6-ko cells (bottom), both treated with control or TRIM37 siRNAs, before immunostaining for CP110 and Centrin-2. (C–D) HeLa cells grown with centrinone for 8 days (C) or RPE-1 HsSAS-6-ko cells (D), both treated with control or TRIM37 siRNAs, before immunostaining for Centrobin and Centrin-2. (E) HeLa cells were synchronized with a double thymidine block, released and transfected with control, TRIM37, Centrobin, or both TRIM37 and Centrobin siRNAs, as indicated. Additionally, DMSO or BI-2536 was added to the cells, which were fixed at time 0 hr or 8 hr after release, before immunostaining with antibodies against CP110 and Centrobin. The percentage of cells with extra CP110 foci was quantified in each condition. Chart shows the average and SDs from three independent experiments (n = 50 cells each). (F) HeLa cells were synchronized with a double thymidine block, released and transfected with control, TRIM37, Centrobin, or both TRIM37 and Centrobin siRNAs, as indicated. Moreover, DMSO or BI-2536 was added to the cells, which were fixed at time 0 hr or 8 hr after release, before immunostaining with antibodies against CP110 and HsSAS-6. (G) Corresponding percentage of cells with extra CP110 foci, with an indication of the fraction of them bearing HsSAS-6. Chart shows the average and SDs from two independent experiments (n = 50 cells each). (H) Working model of TRIM37 role in preventing formation of supernumerary MTOCs. Our findings lead us to propose that TRIM37 prevents the formation of supernumerary Centrin foci through two pathways mediated by Centrobin (top) and PLK1 (bottom). The Centrobin pathway relies on tiger Centrobin assemblies that act as platforms for PLK4-dependent Cenpas formation. Thereafter, Cenpas could evolve into centriole-related structures with the stepwise incorporation of other centriolar proteins such as HsSAS-6. We propose that the PLK1 pathway might reflect its role in promoting centriole disengagement. Note that only extra MTOCs are represented. See text for details. Source data for panels A, E, and G can be found in *Figure 6—source data 1*.

The online version of this article includes the following source data and figure supplement(s) for figure 6:

**Source data 1.** Source data for figure panels: *Figure 6A, E and G* and *Figure 6—figure supplement 1A*.

**Figure supplement 1.** PLK4 activity and HsSAS-6 are dispensable for Centrobin assemblies.

Further evidence supporting the existence of two parallel pathways was obtained by examining the distribution of HsSAS-6 in cells depleted of TRIM37 plus either PLK1 or Centrobin. Indeed, we found that Cenpas generated upon combined TRIM37 depletion and PLK1 inhibition, which thus rely strictly on Centrobin, rarely harbored HsSAS-6 (*Figure 6F,G*). By contrast, Cenpas generated upon double depletion of TRIM37 and Centrobin, which thus rely strictly on PKL1, frequently harbored HsSAS-6 (*Figure 6F,G*). Taken together, our findings indicate that two pathways are triggered when TRIM37 is lacking: one that relies on elongated Centrobin assemblies that act as a platform to assemble Cenpas, which at the least is initially independent of HsSAS-6, and another one mediated by PLK1 that operates through HsSAS-6 recruitment (*Figure 6H*, see Discussion).

## Discussion

Centriole number control is critical for proper cell physiology, including genome integrity. Assemblies of centriolar proteins that can recruit PCM and nucleate microtubules despite not being bona fide centrioles must likewise be kept in check. Here, we identify the TRIM37 E3 ligase, which is mutated in Mulibrey nanism, as a critical component that prevents the formation of centriolar protein assemblies (Cenpas) through two independent pathways relying on Centrobin and PLK1. Of particular interest, we uncover that TRIM37 depletion results in the formation of striated Centrobin assemblies that we propose serve as platforms for Cenpas generation.

### Two pathways together result in Cenpas upon TRIM37 depletion

What mechanisms lead to Cenpas formation upon TRIM37 depletion? We previously hypothesized that TRIM37 could act by restricting centriole reduplication in G2, since PLK1 inhibition in TRIM37-depleted cells reduced Cenpas formation (*Balestra et al., 2013*). However, some Cenpas remained upon such inhibition. Moreover, while Cenpas formed upon TRIM37 depletion as early as 4 hr after the G1/S transition (*Balestra et al., 2013*), PLK1-mediated centriole reduplication occurs only 24 hr after G2 arrest (*Loncarek et al., 2010*). Here, we obtained further evidence that Cenpas do not form solely through a reduplication mechanism. First, some Cenpas appear away from resident centrioles. Second, most Cenpas do not harbor the procentriolar protein HsSAS-6, at the least initially, in contrast to the situation during centriole reduplication during G2 arrest (*Loncarek et al., 2010*). Furthermore, analysis with CLEM revealed that Cenpas are usually either centriole-related structures or novel striped electron-dense structures, and not bona fide procentrioles. Together, these findings indicate that Cenpas do not form solely through centriole reduplication, but also through an alternative de novo pathway.

Our findings indicate that this alternative pathway relies on Centrobin, since the joint removal of PLK1 and Centrobin entirely prevent Cenpas generation. Centrobin is a coiled-coil containing protein that contributes to several aspects of centriole assembly and growth, as well as to ciliogenesis (*Gudi et al., 2011*; *Ogungbenro et al., 2018*; *Zou et al., 2005*). Centrobin can stabilize and promote microtubule nucleation (*Gudi et al., 2011*; *Jeong et al., 2007*; *Shin et al., 2015*), and whether this property is important for Cenpas formation upon TRIM37 depletion will be interesting to test.

The heterogeneity in Cenpas ultrastructure uncovered by CLEM might also reflect the co-existence of these two independent assembly pathways. In addition, such heterogeneity may reflect a step-wise generation of Cenpas. This possibility is compatible with the fact that, compared to the 48 hr post-transfection analyzed here, a higher number of extra HsSAS-6 foci are present 72 hr after transfection with TRIM37 siRNAs (*Balestra et al., 2013*). Therefore, HsSAS-6 might not be present or required initially for de novo Cenpas formation, but contribute later to their consolidation. Although HsSAS-6-ko cells depleted of TRIM37 can assemble some Cenpas, perhaps they are more rudimentary ones. Regardless, de novo Cenpas generation upon TRIM37 depletion must in some way differ from the classical de novo centriole assembly, since that pathway is fully reliant on HsSAS-6 (*Wang et al., 2015*).

## On the relationship of Centrobin and PLK4 upon TRIM37 depletion

PLK4 is essential for forming all Cenpas upon TRIM37 depletion, as it is essential for centriole reduplication and classical de novo centriole assembly (*Habedanck et al., 2005*; *Wong et al., 2015*). How could PLK4 be required for Cenpas generation stemming from the Centrobin assemblies formed upon TRIM37 depletion? We found that the PLK4 condensates previously observed in TRIM37-ko cells (*Meitinger et al., 2020*; *Meitinger et al., 2016*) coincide with the Centrobin assemblies uncovered here. Intriguingly, PLK4 localization at Centrobin assemblies was detected with only one out of three antibodies tested (*Moyer et al., 2015*; *Sillibourne et al., 2010*; *Wong et al., 2015*). We speculate that PLK4 might be poorly accessible at Centrobin assemblies for the two other antibodies (see Materials and methods). Interestingly, while Centrobin depletion precluded formation of PLK4 condensates, PLK4-depleted cells still harbored Centrobin structures. Intriguingly, PLK4 depletion did not diminish the signal detected by those PLK4 antibodies that showed colocalization with elongated Centrobin assemblies (*Wong et al., 2015*). Perhaps PLK4 exhibits low turnover within Centrobin assemblies, such that it remains present in that location despite siRNA-mediated depletion. Alternatively, the signal recognized in the Centrobin assemblies by these PLK4 antibodies might not be specific. In any case, it is interesting to note that in *Xenopus* extracts PLK4 self-assembles into condensates that recruit γ-tubulin and behave as MTOCs (*Montenegro Gouveia et al., 2018*), raising the possibility that Centrobin assemblies formed upon TRIM37 depletion may serve as platforms to recruit such condensates.

## Cenpas threaten cell physiology

Centriolar protein assemblies have been reported in other contexts, although with different molecular origins (*Li et al., 2012*; *Shiratsuchi et al., 2015*). Thus, Neurl4 interacts with CP110 and promotes its destabilization, such that Neurl4 depletion results in increased CP110 protein levels and ectopic MTOCs (*Li et al., 2012*). Likewise, RMB14 limits formation of the STIL/CPAP complex, with RMB14 depletion triggering formation of centriolar protein complexes that do not initially require HsSAS-6 for their assembly and results in ectopic MTOCs (*Shiratsuchi et al., 2015*). Centrobin distribution was inspected upon both Neurl14 and RMB14 depletion, and no elongated structures like the ones reported here were reported (*Li et al., 2012*; *Shiratsuchi et al., 2015*), suggesting that different assembly routes operate in those cases. Although these previously reported centriolar protein assemblies and the ones analyzed here do not share a clear common molecular composition or assembly route, we propose to group them jointly under the acronym Cenpas, reflecting the fact that they all form following some de novo process, resulting in the generation of centriole-related structures that behave as active MTOCs.

To our knowledge, our findings represent the first example in which Cenpas have been reported in a human genetic disorder. The fact that Cenpas are present in Mulibrey derived patient cells raises the possibility that some disease features could be due to Cenpas formation, perhaps owing to the extra MTOCs and resulting chromosome miss-segregation phenotype. As one of the characteristics

of Mulibrey nanism is a propensity to develop tumors, we speculate that the presence of Cenpas could contribute to this phenotype, since extra centrioles can promote tumorigenesis (*Ganem et al., 2009*; *Godinho et al., 2014*; *Levine et al., 2017*; *Serçin et al., 2016*). By extenstion, it will be interesting to investigate whether some of the instances in which extra centriole numbers are observed in solid and hematological tumors may in reality correspond to Cenpas.

# Materials and methods

## Key resources table

| Reagent type (species) or resource | Designation | Source or reference | Identifiers | Additional information |
|---|---|---|---|---|
| Gene (*Homo sapiens*) | TRIM37 | Ensembl | ENST00000 262294.12 | |
| Cell line (*Homo sapiens*) | HeLa Kyoto | *Schmitz and Gerlich, 2009* | | |
| Cell line (*Homo sapiens*) | HeLa Centrin-1:GFP | *Piel et al., 2000* | | |
| Cell line (*Homo sapiens*) | U2OS | Sigma | 92022711 | |
| Cell line (*Homo sapiens*) | hTERT-RPE-1 | ATCC | CRL-4000 | |
| Cell line (*Homo sapiens*) | hTERT-RPE-1 p53 -/- Centrin-1:eGFP | This work | | carrying an integrated plasmid (pCW57.1) expressing Centrin-1:eGFP under a doxycycline inducible promoter (generous gift from George Hatzopoulos) |
| Cell line (*Homo sapiens*) | hTERT-RPE-1 Centrobin -/- | *Ogungbenro et al., 2018* | | |
| Cell line (*Homo sapiens*) | hTERT-RPE-1 TRIM37 -/- | *Meitinger et al., 2016* | | |
| Cell line (*Homo sapiens*) | primary fibroblast | This work | | skin biopsy samples a control individual, with approval by the Institutional Review Board of the Helsinki University Central Hospital (183/13/03/03/2009). |
| Cell line (*Homo sapiens*) | primary fibroblast | This work | | skin biopsy samples from a Mulibrey nanism patient (P-1) homozygous for the Finnish founder mutation with approval by the Institutional Review Board of the Helsinki University Central Hospital (183/13/03/03/2009). |
| Cell line (*Homo sapiens*) | primary fibroblast | This work | | skin biopsy samples from a Mulibrey nanism patient (P-2) homozygous for the Finnish founder mutation with approval by the Institutional Review Board of the Helsinki University Central Hospital (183/13/03/03/2009). |
| Antibody | Anti-TRIM37 (Rabbit polyclonal) | Bethyl Laboratories | A301-174A | WB (1/1000) |
| Antibody | Anti- α-tubulin (Mouse monoclonal) | Sigma-Aldrich | T6199 (DM1a) | WB (1/13000) IF (1/1000) |
| Antibody | Anti-Centrobin (Rabbit polyclonal) | Atlas Ab | HPA023321 | WB (1/500) IF (1/1000) |
| Antibody | Anti-Centrobin (Mouse monoclonal) | Abcam | Ab70448 | IF (1/1000) |
| Antibody | Anti-HSP70 (Mouse monoclonal) | Santa Cruz Biotechnology | sc-24 | WB (1/20000) |
| Antibody | Anti-GFP (human) | Antibody platform of Institute Curie | A-R-H#11 | IF (1/50) |
| Antibody | Anti- α-tubulin (human) | Antibody platform of Institute Curie | A-R-H#02 | IF (1/50) |

*Continued on next page*

*Continued*

| Reagent type (species) or resource | Designation | Source or reference | Identifiers | Additional information |
|---|---|---|---|---|
| Antibody | Anti-HsSAS-6 (Mouse monoclonal) | Santa Cruz Biotechnology | sc-81431 | IF (1/500) |
| Antibody | Anti-CEP63 (Rabbit polyclonal) | Millipore | 06–1292 | IF (1/1000) |
| Antibody | Anti-CEP152 (Rabbit polyclonal) | Sigma Aldrich | HPA039408 | IF (1/2000) |
| Antibody | Anti-acetylated tubulin (Mouse monoclonal) | Sigma Aldrich | T6793 | IF (1/1000) |
| Antibody | Anti-γ-tubulin (Mouse monoclonal) | Sigma Aldrich | T5326 | IF (1/1000) |
| Antibody | Anti-Centrin 2 (Mouse Monoclonal) | Sigma Aldrich | 20H5 | IF (1/1000) |
| Antibody | Anti-CEP164 (Rabbit polyclonal) | Novus Biologicals | 45330002 | IF (1/1000) |
| Antibody | Anti-CP110 (Rabbit polyclonal) | Proteintech | 12780–1-AP | IF (1/1000) |
| Antibody | Anti-CEP135 (Rabbit polyclonal) | Abcam | Ab-75005 | IF (1/1000) |
| Antibody | Anti-CPAP (Rabbit polyclonal) | *Kohlmaier et al., 2009* | | IF (1/500) |
| Antibody | Anti-SPICE (Rabbit polyclonal) | Sigma Aldrich | HPA064843 | IF (1/500) |
| Antibody | Anti-Ninein (Rabbit polyclonal) | *Delgehyr et al., 2005* | L77 | IF (1/8000) |
| Antibody | Anti-hPOC5 (Rabbit polyclonal) | *Azimzadeh et al., 2009* | | IF (1/1000) |
| Antibody | Anti-C-Nap (Mouse monoclonal) | BD Biosciences | 611374 | IF (1/400) |
| Antibody | Anti-STIL (Rabbit polyclonal) | Abcam | Ab-222838 | IF (1/2000) |
| Antibody | Anti-PCNT (Rabbit polyclonal) | Abcam | Ab-4448 | IF (1/1000) |
| Antibody | Anti-AKAP450 (Mouse monoclonal) | BD Biosciences | 611518 | IF (1/1000) |
| Antibody | Anti-CDK5-Rap2 (Rabbit polyclonal) | Millipore | 06–1398 | IF (1/1000) |
| Antibody | Anti-CEP192 (Rabbit polyclonal) | *Zhu et al., 2008* | | IF (1/1000) |
| Antibody | Anti-CEP170 (Rabbit polyclonal) | Sigma Aldrich | HPA042151 | IF (1/1000) |
| Antibody | Anti-P-T210-PLK1 (Mouse monoclonal) | BD Bioscience | 558400 | IF (1/1000) |
| Antibody | Anti-Rootletin (Rabbit polyclonal) | Santa Cruz Biotechnology | sc-374056 | IF (1/200) |
| Antibody | Anti-CEP68 (Rabbit polyclonal) | Atlas Ab | HPA040493 | IF (1/1000) |
| Antibody | Anti-PLK4 (Rabbit polyclonal) | *Sillibourne et al., 2010* | raised against PLK4 kinase domain residues 13–265 | IF (1/1000) |
| Antibody | Anti-PLK4 (Rabbit polyclonal) | *Moyer et al., 2015* | raised against residues 510-970 | IF (1/1000) |
| Antibody | Anti-PLK4 (Rabbit polyclonal) | *Wong et al., 2015* | raised against residues 814–970 | IF (1/2000) |

*Continued on next page*

*Continued*

| Reagent type (species) or resource | Designation | Source or reference | Identifiers | Additional information |
|---|---|---|---|---|
| Recombinant DNA reagent | pEBTet-TRIM37:GFP (plasmid) | *Balestra et al., 2013* | | GFP fused version of TRIM37 into pEB-tet plasmid under a doxycycline inducible promoter |
| Recombinant DNA reagent | p-Centrobin:GFP (plasmid) | *Shin et al., 2015* | | GFP fused version of Centrobin |
| Recombinant DNA reagent | pEGFP:SPICE (plasmid) | *Archinti et al., 2010* | | GFP fused version of SPICE |
| Recombinant DNA reagent | pcDNA3-TRIM37:GFP (plasmid) | This paper | | GFP fused version of TRIM37 |
| Recombinant DNA reagent | pcDNA3-TRIM37: NES:GFP (plasmid) | This paper | | GFP fused version of TRIM37 with the HIV-Rev NES sequence (LQLPPLERLTLD) *Wen et al., 1995* |
| Sequenced-based reagent | CNTROB-FW | This paper | PCR primers | 5′-GTCTCCATCTAGCTCAGCCC-3′ |
| Sequenced-based reagent | CNTROB-RV | This paper | PCR primers | 5′-AGGCTCTGAATATGGCGCT C-3′ |
| Sequenced-based reagent | TRIM37-FW | This paper | PCR primers | 5′-TGCCATCTTACGATTCAGCTAC-3′ |
| Sequenced-based reagent | TRIM37- RV | This paper | PCR primers | 5′-CGCACAACTCCATTTCCATC-3′ |
| Sequenced-based reagent | GAPDH-FW | This paper | PCR primers | 5′-GGAAGGTGA AGGTCGGAGTC-3′ |
| Sequenced-based reagent | GAPDH-RV | This paper | PCR primers | 5′-GTTGAGGTCAATGAAGGGGTC-3′ |
| Sequenced-based reagent | siRNA-TRIM37 | Invitrogen *Balestra et al., 2013* | | 5′-UUAAGGACCGGA GCAGUAUAGAAAA-3′ |
| Sequenced-based reagent | siRNA-Centrobin | Invitrogen *Zou et al., 2005* | | 5′-AGUGCCAGACUGCAGCAACGGGAAA-3′ |
| Sequenced-based reagent | siRNA-SPICE | Invitrogen *Archinti et al., 2010* | | 5′-GCAGCUGAGAACAAAUGAGUCAUUA-3′ |
| Sequenced-based reagent | siRNA-HsSAS-6 | Invitrogen *Strnad et al., 2007* | | 5′-GCACGUUAAUCAGCUACAAUU-3′ |
| Sequenced-based reagent | siRNA-STIL | Invitrogen *Kitagawa et al., 2011a* | | 5′-AACGUUUACCAUACAAAGAAA-3′ |
| Sequenced-based reagent | siRNA-CPAP | Invitrogen *Kitagawa et al., 2011a* | | 5′-AGAAUUAGCUCGAAUAGAA-3′ |
| Sequenced-based reagent | siRNA-PLK4 | Invitrogen *Balestra et al., 2013* | | 5′-GAAAUGAACAGGUAUCUAA-3′ |
| Sequenced-based reagent | Stealth RNAi siRNA Negative Control Lo GC | Invitrogen | 12935200 | |
| Chemical compound | BI-2536 | Selleck Chemicals | S1109 | 10 µM |
| Chemical compound | RO-3306 | Sigma Aldrich | SML0569 | 10 µM |
| Chemical compound | Centrinone | MCE | Hy-18682 | 125 nM |
| Chemical compound | Cycloheximide | Sigma Aldrich | C7698 | 150 µg/ml |

## Cell culture, cell lines, and cell treatments

HeLa Kyoto (Cellosaurus ref: CVCL_1922, *Schmitz and Gerlich, 2009*, generous gift from Daniel Gerlich, Institute of Molecular Biotechnology, Vienna, Austria) and U2OS (Cellosaurus ref: CVCL_0042, provided by Sigma, 92022711) cells were grown in high glucose DMEM medium (Sigma-Aldrich), hTERT-RPE-1 (Cellosaurus ref: CVCL_4388, provided by ATCC CRL-4000) cells in high glucose DMEM/F-12 medium (Sigma-Aldrich). Fibroblast cultures were established from skin biopsy samples of two Mulibrey nanism patients homozygous for the Finnish founder mutation, as well as a control individual, with approval by the Institutional Review Board of the Helsinki University Central Hospital (183/13/03/03/2009). The patients signed an informed consent for the use of fibroblast cultures. Other cell lines used were HeLa cells carrying an integrated plasmid expressing Centrin-1:GFP (*Piel et al., 2000*) where the expected Centrin-1:GFP signal localizing to centrioles was

observed, RPE-1 p53 -/- cells carrying an integrated plasmid (pCW57.1) expressing Centrin-1:eGFP under a doxycycline inducible promoter (reported for the first time in this work, generous gift from George Hatzopoulos), RPE-1 p53 -/- Centrobin knock out cells, confirmed by the lack of Centrobin signal in IF experiments, (*Ogungbenro et al., 2018*) (generous gift from Ciaran Morrison), RPE-1 p53 -/- HsSAS-6 knock out cells, confirmed by the lack of HsSAS-6 signal in IF experiments, (*Wang et al., 2015*) (generous gift from Bryan Tsou) and RPE-1 TRIM37 knock out cells, confirmed by the reported phenotype previously reported by *Meitinger et al., 2016* (generous gift from Karen Oegema). Media were supplemented with 10% fetal bovine serum, with the exception of Mulibrey nanism patient cell lines medium, which was supplemented with 15% fetal bovine serum. In addition, all media were supplemented with 2 mM L-glutamine, 100 units/ml penicillin, and 100 µg/ml strepto-mycin (all from Sigma-Aldrich) and grown at 37°C in 5% $CO_2$. All cell lines tested negative for myco-plasma contamination. HeLa Kyoto cells were synchronized using a double-thymidine block and release protocol as follows: cells were incubated in medium with 2 mM thymidine (Sigma Aldrich, T9250) for 17 hr, released for 8 hr and again incubated with 2 mM Thymidine for 17 hr. For single transfection experiments, control or TRIM37 siRNAs transfections were performed during the 8 hr period between the two thymidine treatments. For double transfection experiments, in addition to the above, either control or Centrobin siRNAs were transfected before the first Thymidine treat-ment. Drugs used in this work were 10 µM BI-2536 (S1109, Selleck Chemicals), 10 µM RO-3306 (Sigma-Aldrich, SML0569), 125 nM Centrinone (MCE, Hy-18682), and 150 µg/ml Cycloheximide (Sigma-Aldrich, C7698).

## Transfections, plasmids, and siRNAs

For siRNA treatments, cells were typically transfected in a six well plate format with 20 µM siRNAs and 4 µL Lipofectamine RNAiMAX (Thermo Fisher Scientific); the depletion phenotype was inspected 72 hr after transfection unless otherwise indicated in the text or the legends. siRNAs sequences were as follows: TRIM37 (5'-UUAAGGACCGGA GCAGUAUAGAAAA-3') (*Balestra et al., 2013*) Cen-trobin (5'-AGUGCCAGACUGCAGCAACGGGAAA-3') (*Zou et al., 2005*), SPICE (5'-GCAGCUGA-GAACAAAUGAGUCAUUA-3') (*Archinti et al., 2010*), HsSAS6 (5'-GCACGUUAAUCAGCUACAAUU-3') (*Strnad et al., 2007*), STIL (5'-AACGUUUACCAUACAAAGAAA-3') (*Kitagawa et al., 2011a*), CPAP (5'-AGAAUUAGCUCGAAUAGAA-3') (*Kitagawa et al., 2011a*), PLK4 ( 5'-GAAAUGAACAGG UAUCUAA-3') (*Balestra et al., 2013*) and Stealth RNAi siRNA Negative Control Lo GC (Ref: 12935200; Invitrogen). For plasmid transient transfection, FuGENE 6 Transfection Reagent (Prom-ega) was used according to the manufacturer's protocol and the phenotype inspected 24 or 48 hr after transfection. Transfected plasmids were as follows: pEBTet-TRIM37:GFP (*Balestra et al., 2013*), pGFP-Centrobin:GFP (pGFP-NIP2) (*Shin et al., 2015*; generous gift from Kunsoo Rhee, Seoul National University, Korea) pEGFP:SPICE (*Archinti et al., 2010*; generous gift from Jens Lüders, IRB, Barcelona, Spain) pcDNA3-TRIM37:GFP and pcDNA3-TRIM37:NES:GFP were generated by cloning the TRIM37 ORF (964 aa) fused to GFP, or to the HIV-Rev NES sequence (LQLPPLERLTLD; *Wen et al., 1995*) and GFP.

## Immunoblotting and cycloheximide chase assay

For western blot analysis, cells were lysed either in 2× Laemmli buffer (4% SDS, 20% glycerol, 125 mM Tris-HCl, pH 6.8) and passed 10 times through a 0.5 mm needle–mounted syringe to reduce vis-cosity, or in NP40 lysis buffer 10 mM Tris/HCl (pH 7.4)/150 mM NaCl/10% (v/v) glycerol/1% (v/v) Nonidet P40/1 mM PMSF and 1 µg/ml of each pepstatin, leupeptin, and aprotinin (Sigma-Aldrich) for 20 min at 4°C and then for 3 min at 37°C, before centrifugation at 20,000 g for 20 min. In this manner, the soluble fraction was separated from the insoluble pellet, which was then solubilized in 1x Laemmli buffer. Lysates were resolved by SDS-PAGE on a 10% polyacrylamide gel and immuno-blotted on Immobilon-P transfer membrane (IPVH00010; 21 Millipore Corporation). Membranes were first blocked with TBS containing 0.05% Tween-20 (TBST) and 5% non-fat dry milk (TBST-5% milk) for 1 hr at room temperature, and then incubated with primary antibodies diluted in PBST-5% milk. Primary antibodies were 1:1000 rabbit anti-TRIM37 (A301-174A; Bethyl Laboratories), 1:30,000 mouse anti-α-tubulin (T6199; Sigma-Aldrich), 1:500 rabbit anti-Centrobin (HPA023321; Atlas), and 1:20,000 mouse anti-HSP70 (sc-24; Santa Cruz). Membranes were washed and incubated for 1 hr in secondary antibodies prepared also in TBST-5% milk. Secondary antibodies were 1:5000 HRP-

conjugated anti-rabbit (W4011; Promega) or mouse (W4021; Promega) IgGs. The signal was detected by standard chemiluminescence (34077; Thermo Scientific). Alternatively, polyacrylamide gels were immunoblotted on low fluorescence PVDF membranes (Immobilon-FL, Millipore), membranes blocked with Odyssey Blocking Buffer (LI-COR) and blotted with appropriate primary antibodies and 1:5000 secondary antibodies IRDye 680RD anti-mouse IgG (H+L) Goat LI-COR (926–68070) and IRDye 800CW anti-rabbit IgG (H+L) Goat LI-COR (926-32211). Membranes were then air-dried in the dark and scanned in an Odyssey Infrared Imaging System (LI-COR), and images analyzed with ImageStudio software (LI-COR). In all cases, membrane washes were in TBST. For the cycloheximide chase experiments, HeLa Kyoto cells were treated with fresh DMEM containing 150 µg/ml cycloheximide (CHX). Cells were collected 0, 2, 4, 6, and 8 hr after CHX addition, and protein extracts prepared in 2× Laemmli buffer as described above. A total of 40 µg of siControl lysate and 20 µg of siTRIM37 lysate were resolved by SDS-PAGE, analyzed by immunoblotting with Centrobin and α-tubulin antibodies before quantification with ImageStudio. The siControl and siTRIM37 conditions at time 0 were normalized as 100%, and the other conditions for the same siRNA treatment expressed relative to this. Centrobin expression was quantified as the Centrobin signal divided by the α-tubulin signal.

## RNA isolation, reverse transcription, and real-time PCR

RNA was extracted using the RNeasy Mini kit according to the manufacturer's instruction (QIAGEN), including DNase I to avoid potential contaminations with DNA. 3 µg of total RNA, random hexamers and SuperScript III Reverse Transcriptase (InvitrogenTM) were used to obtain complementary DNA (cDNA). Quantitative PCR from cDNA was performed to assess siRNA-mediated knock-down of TRIM37 and Centrobin, using iTaq Universal SYBR Green Supermix following the manufacturer's instructions (Bio-Rad) in an Applied Biosystems 7500 Fast Real-time PCR System (Thermo Fisher Scientific). Relative mRNA levels of the indicated genes were calculated by the 2-DDCT method (Bulletin 5279, Real-Time PCR Applications Guide, Bio-Rad), using GAPDH expression as endogenous control. The primer sequences used were: Centrobin:_CNTROB-FW 5'-GTCTCCATCTAGCTCAGCCC-3', CNTROB-RV 5'-AGGCTCTGAATATGGCGCT C-3', TRIM37: TRIM37-FW 5'-TGCCATCTTACGATTCAGCTAC-3', TRIM37-RV 5'-CGCACAACTCCATTTCCATC-3'. GAPDH: GAPDH-FW 5'-GGAAGGTGA AGGTCGGAGTC-3', GAPDH-RV 5'-GTTGAGGTCAATGAAGGGGTC-3'.

## Cell cycle analysis

Cells were fixed with cold 70% ethanol overnight at 4°C, incubated with PBS containing 250 µg/mL RNase A (Sigma) and 10 µg/mL propidium iodide (Fluka) at room temperature for 30 min, before analysis with a FACSCalibur Flow Cytometer (BD). Cell cycle distribution data were further analyzed using ModFit LT 3.0 software (Verity Software House Inc).

## Indirect immunofluorescence and microtubule-regrowth assay

Cells were grown on glass coverslips and fixed for 7 min in −20°C methanol, washed in PBS, and blocked for 30 min in PBS 0.05% Tween 20 (PBST) with 1% bovine serum albumin. Cells were incubated overnight at 4°C with primary antibodies, washed three times for 5 min with PBST, incubated for 1 hr at room temperature with secondary antibodies, washed three times for 5 min in PBST and mounted in Vectashield mounting medium with DAPI (H-1200; Vector Laboratories). Primary antibodies used for immunofluorescence were: 1:50 human anti-GFP (hVHH antiGFP-hFc, A-R-H#11) and human anti α-tubulin (A-R-H#02) from the recombinant antibody platform of Institut Curie, 1:1000 rabbit anti-GFP (RGFP-45ALY-Z; ICL), 1:500 mouse anti-HsSAS-6 (sc-81431; Santa Cruz), 1:1000 rabbit anti-CEP63 (06–1292; Millipore), 1:2000 rabbit anti-CEP152 (HPA039408; Sigma-Aldrich), 1:1000 mouse anti-acetylated tubulin (T6793; Sigma-Aldrich), 1:1000 mouse anti-γ-tubulin (GTU88, T5326; Sigma-Aldrich), 1:1000 mouse anti-Centrin2 (20H5; Sigma-Aldrich), 1:2000 rabbit anti-CEP164 (45330002; Novus Biologicals), 1;1000 mouse anti-α-tubulin (T6199; Sigma-Aldrich), 1:1000 rabbit anti-CP110 (12780–1-AP; Proteintech), 1:1000 mouse anti-Centrobin (ab70448; Abcam), 1:1000 rabbit anti-Centrobin (HPA023321; Atlas Antibodies), 1:1000 rabbit anti-CEP135 (ab75005; Abcam), 1:500 rabbit anti-CPAP (*Kohlmaier et al., 2009*), 1:500 rabbit anti-SPICE (HPA064843, Sigma-Aldrich), 1:8000 rabbit anti-Ninein (L77, *Delgehyr et al., 2005*), 1:1000 rabbit anti-hPOC5 (*Azimzadeh et al., 2009*) (a generous gift from Michel Bornens), 1:400 mouse anti-C-Nap (611374;

BD Biosciences) 1:2000 rabbit anti-STIL (ab222838; Abcam), 1:1000 rabbit anti-PCNT (ab4448; Abcam), 1:1000 mouse anti-AKAP450 (611518; BD Biosciences), 1:1000 rabbit anti-CDK5Rap2 (06–1398; Millipore), 1:1000 rabbit anti-CEP192 (*Zhu et al., 2008*)(a generous gift from Laurence Pelletier), 1:1000 rabbit anti-CEP170 (HPA042151; Sigma-Aldrich), 1:1000 mouse anti-P-T210-PLK1 (558400; BD Bioscience), 1:200 mouse anti-Rootletin (sc-374056, Santa Cruz Biotechnology), 1:1000 rabbit anti-CEP68 (HPA040493, Atlas Antibodies). We also utilized three PLK4 antibodies: 1:1000 rabbit anti-PLK4(KD) (*Sillibourne et al., 2010*; generous gift from Michel Bornens), raised against residues 13–265; 1:1000 rabbit anti-PLK4 (*Moyer et al., 2015*; a generous gift from Andrew Holland), raised against residues 510–970; as well as 1:2000 rabbit anti PLK4 (*Wong et al., 2015*; a generous gift from Karen Oegema), raised against residues 814–970. Secondary antibodies were 1:500 mouse Alexa-488, 1:3000 rabbit Cy3, 1:3000 human Alexa-633, 1:1000 mouse Alexa-649, and 1:500 human Alexa-488, all from Jackson ImmunoResearch. For microtubule depolymerization-regrowth experiments, cells were first incubated at 4°C for 30 min, then rinsed in pre-warmed medium (37°C), followed by incubation at room temperature for 1–2 min to allow microtubule regrowth. Thereafter, cells were fixed and stained as described above. Image J was used to measure PLK4 signal within Centrobin structures. A mask was generated using Centrobin staining to determine the area of the structure and average PLK4 intensity within the area was registered.

## Live imaging, ultrastructure expansion microscopy, and confocal microscopy

HeLa Centrin-1:GFP cells were transfected with control or TRIM37 siRNAs for 48 hr, transferred to 35 mm imaging dishes (Ibidi, cat.no 81156), and imaged at 37°C and 5% CO2 in medium supplemented with 25 mM HEPES (Thermofisher) and 1% PenStrep (Thermofisher). Combined DIC and GFP-epifluorescence time-lapse microscopy was performed on a motorized Zeiss Axio Observer D1 using a 63 × 1.4 NA plan-Apochromat oil immersion objective, equipped with an Andor Zyla 4.2 sCMOS camera, a piezo controlled Z-stage (Ludl Electronic Products), and an LED light source (Lumencor SOLA II). Imaging was conducted every 10 min, capturing Z-stacks of optical sections 0.5 µm apart, covering a total height of 8 µm. Ultrastructure expansion microscopy was conducted essential as reported (*Gambarotto et al., 2019*). For imaging, the sample was mounted on a 25 mm round poly-D-lysine coated precision coverslip. STED imaging was performed on a Leica TCS SP8 STED 3X microscope with a 100 × 1.4 NA oil-immersion objective. Secondary antibodies were 1:500 Alexa-488 (A-11039; Thermofisher), Alexa-594 (ab150072; Abcam), and Atto647N (2418; Hypermol). Confocal images were captured on a Leica TCS SP5 with a HCX PL APO Lambda blue 63 × 1.4 NA oil objective. All images shown are maximal intensity projections of relevant planes. Image processing was carried out using Image J and Adobe Photoshop (Adobe).

## Correlative light electron microcopy (CLEM)

HeLa and RPE-1 cells expressing Centrin-1:GFP were cultured in glass-bottom Petri dishes (MatTek, Cat. No. P35G-1.5–14-CGRD), with an alpha-numeric grid pattern, and transfected with control or TRIM37 siRNAs. Cells were chemically fixed 72 hr after transfection with a buffered solution of 1% glutaraldehyde 2% paraformaldehyde in 0.1 M phosphate buffer at pH 7.4. Dishes were then screened with a wide-field fluorescent microscope (Zeiss Observer D1, using a 63 × 1.4 NA oil objective) to identify cells of interest, which were imaged with both transmitted and fluorescence microscopy to register the position of each cell on the grid, as well as the location of their GFP foci, capturing optical slices 500 nm apart. The cells were then washed thoroughly with cacodylate buffer (0.1M, pH 7.4), postfixed for 40 min in 1.0% osmium tetroxide 1.5% potassium ferrocyanide, and then for 40 min in 1.0% osmium tetroxide alone. Finally, cells were stained for 40 min in 1% uranyl acetate in water before dehydration through increasing concentrations of alcohol and then embedding in Durcupan ACM resin (Fluka, Switzerland). The coverslips were then covered with 1 mm of resin, which was hardened for 18 hr in a 65°C oven. The coverslips were removed from the cured resin by immersing them alternately into 60°C water followed by liquid nitrogen until the coverslips parted. Regions of resin containing the cells of interest were then identified according to their position on the alpha-numeric grid, cut away from the rest of the material and glued to blank resin block. Ultra-thin (50 nm thick) serial sections were cut through the entire cell with a diamond knife (Diatome) and ultramicrotome (Leica Microsystems, UC7), and collected onto single slot grids with a

pioloform support film. Sections were further contrasted with lead citrate and uranyl acetate and images taken in a transmission electron microscope (FEI Company, Tecnai Spirit) operating at 80 kV, with a digital camera (FEI Company, Eagle). To correlate the light microscopy images with the EM images and identify the exact position of the Centrin-1:GFP foci, fluorescent images were overlaid onto the electron micrographs of the same cell using Photoshop.

## Statistical analysis

Statistical significance was determined with an unpaired Student's t-test using PRISM software (Graphpad Software Inc). See *Supplementary file 2* for exact values.

## Acknowledgements

We thank Graham Knott and Marie Croisier (BioEM platform of the School of Life Sciences, EPFL) for assistance with CLEM and TEM, Isabelle Flückiger for technical support in the early stages of the project, as well as Niccolò Banterle for advice with U-ExM. Anna-Elina Lehesjoki (Folhälsan Research Center, Helsinki and University of Helsinki, Helsinki, Finland) is acknowledged for her contribution towards securing patient and control fibroblasts, Daniel Gerlich (Vienna BioCenter, Austria), Ciaran Morrison (Centre for Chromosome Biology, Galway, Ireland), Bryan Tsou (Memorial Sloan Kettering Cancer Center, New York, USA), Karen Oegema (University of California San Diego, USA) and George Hatzopoulos (EPFL, Lausanne, Switzerland) for cell lines, Michel Bornens (Institut Curie, Paris, France), Andrew Holland (Johns Hopkins University School of Medicine, Baltimore, USA), Laurence Pelletier (Lunenfeld-Tanenbaum Research Institute, Toronto, Canada) and Karen Oegema for antibodies. Further thanks extend to Kunsoo Rhee, Seoul National University, Korea and Jens Lüders (IRB, Barcelona, Spain) for their gift of plasmids. We are also grateful to Niccolò Banterle and Fernando Monje Casas for critical reading of the manuscript. This work has been supported in part by the Swiss Cancer Research Foundation (KFS-3388-02-2014, to PG). AB was supported by the National Centre for Competence in Research (NCCR) Chemical Biology, funded by the Swiss National Science Foundation. FRB thanks PG for having hosted and supported him at the beginning of the project. FRB was funded by a Marie Curie IEF Postdoctoral Fellowship (PIEF_GA-2013–629414; CSIC-Cabimer, Seville, Spain) and by the University of Seville through a postdoctoral contract of the V PPIT-US (US-Cabimer, Seville, Spain). CABIMER is supported by the regional government of Andalucia (Junta de Andalucía).

## Additional information

### Funding

| Funder | Grant reference number | Author |
| --- | --- | --- |
| Swiss Cancer Research foundation | KLS-3388-02-2014 | Pierre Gönczy |
| European Research Council | Marie Curie Intra-European Fellowships PIEF-GA-2013-629414 | Fernando R Balestra Rosa M Ríos |
| Swiss National Science Foundation | | Alizée Buff |
| University of Seville | postdoctoral contract of the V PPIT-US | Fernando R Balestra |
| Junta de Andalucía | CABIMER | Fernando R Balestra |

The funders had no role in study design, data collection and interpretation, or the decision to submit the work for publication.

### Author contributions

Fernando R Balestra, Conceptualization, Data curation, Formal analysis, Supervision, Funding acquisition, Validation, Investigation, Methodology, Writing - original draft; Andrés Domínguez-Calvo, Benita Wolf, Data curation, Formal analysis, Investigation; Coralie Busso, Investigation,

Methodology; Alizée Buff, Investigation; Tessa Averink, Resources, Investigation; Marita Lipsanen-Nyman, Resources, Methodology; Pablo Huertas, Supervision, Writing - review and editing; Rosa M Ríos, Supervision, Funding acquisition; Pierre Gönczy, Conceptualization, Resources, Formal analysis, Supervision, Funding acquisition, Investigation, Writing - review and editing

### Author ORCIDs
Fernando R Balestra ⑩ https://orcid.org/0000-0003-2741-6068
Andrés Domínguez-Calvo ⑩ https://orcid.org/0000-0002-3689-227X
Benita Wolf ⑩ http://orcid.org/0000-0001-5673-4239
Pablo Huertas ⑩ http://orcid.org/0000-0002-1756-4449
Pierre Gönczy ⑩ https://orcid.org/0000-0002-6305-6883

### Ethics
Human subjects: Fibroblast cultures were established from skin biopsy samples with approval by the Institutional Review Board of the Helsinki University Central Hospital (183/13/03/03/2009). The patients signed an informed consent for the use of fibroblast cultures.

### Decision letter and Author response
Decision letter https://doi.org/10.7554/eLife.62640.sa1
Author response https://doi.org/10.7554/eLife.62640.sa2

## Additional files

### Supplementary files
• Supplementary file 1. Summary table of CLEM analysis. Summary of CLEM analysis of HeLa or RPE-1 cells depleted of TRIM37, with number of GFP foci, as well as corresponding resident centriole/procentriole and Cenpas ultrastructure identified by CLEM. See main text for further details. Note that no distinct ultrastructure was found for 5 Centrin-1:GFP foci in cell 4, perhaps reflecting a technical issue in this case.

• Supplementary file 2. Statistical analysis table. Table summarizing the statistical analysis show in main Figures and Figures Supplements. Statistical significance was determined with an unpaired Student's t-test using PRISM software (Graphpad Software Inc).

• Transparent reporting form

### Data availability
All data generated or analysed during this study are included in the manuscript and supporting files.

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
