## [Decision Letter]

[Editors' note: this paper was reviewed by Review Commons.]

**Acceptance summary:**

TRIM37 E3 ubiquitin ligase deficiency causes the rare disorder Mulibrey nanism. This work provides first insights into the formation and regulation of centriole-like assemblies that form in cells lacking TRIM37 activity. By functioning as ectopic MTOCs these assemblies can interfere with bipolar mitotic spindle assembly, causing chromosome segregation errors.

---

## [Author Response]

We thank the reviewers for their important comments and suggestions. We have addressed in full the points that have been raised by carrying out additional experiments and revising the manuscript accordingly, as detailed below point-by-point.

Reviewer #1 (Evidence, reproducibility and clarity (Required)):The manuscript describes the formation of supernumerary centriole protein assemblies (cenpas) upon silencing of the E3 ubiquitin ligase TRIM37. These cenpas resemble centrioles, centriole precursors, or electron-dense striped structures, termed tigers. Similar observations are made in cells from patients lacking functional alleles of TRIM37. The cenpas usually lack the full complement of centriolar proteins, but contain increased amounts of the pro-centriole marker centrobin. It is further shown that the formation of cenpas depends on centrobin, or on a parallel pathway involving Plk1 and SAS-6.Overall, the experiments in this study are of high technical quality and most of them are carefully controlled. The discovery of centrobin-containing striped protein assemblies (tigers) is very interesting and provokes the question of their molecular composition and their mechanistic role in centriole assembly. Since striated fibres containing the protein rootletin have a similar periodicity of stripes (75nm) as the tigers in this study (Vlijm et al., PNAS 2018, 115:E2246-53), I was wondering whether the authors couldn't simply test for co-localization of their tiger stripes with rootletin. A potential identity of tigers with striated fibres would help understanding the mechanisms of cenpas and centriole assembly upon depletion of TRIM37: striated fibres or tigers might be controlling the balance of centriole cohesion vs. disengagement and thereby centriole duplication, or they might play a role in the recruitment of additional proteins involved in pro-centriole assembly.

We are grateful to the reviewer for making this interesting suggestion. We investigated this possibility but found that centrobin “tigers” do not contain Rootletin or CEP68. This is reported in the novel Figure 4—figure supplement 1C, D, as well as the accompanying text in the revised manuscript. Note that we also conducted immunofluorescence analysis with PLK4 antibodies, which we found to colocalize with Centrobin “tiger” structures, in line with the requirement of PLK4 activity for the presence of extra Centrin foci. Moreover, we found that Centrobin is needed for the accumulation of PLK4 in such structures; by contrast, PLK4 depletion does not prevent the formation of Centrobin assemblies. These additional findings are reported in Figure 4D, 4E, Figure 4—figure supplement 1H, I, J and in the accompanying text. Note also that only one of the PLK4 antibodies we tested labeled Centrobin condensates, a point that is mentioned explicitly and discussed in the manuscript.

In the same context, did the authors correct for the experimentally induced sample expansion in Figure 5B, when comparing inter-stripe distances between U-ExM and EM samples?

Indeed, this was taken into account. That this was the case is explained explicitly in the revised manuscript.

Other major points:The amount of TRIM37-depletion upon siRNA-treatment should be indicated prominently. I see in the Materials and methods and in Figure S4 that quantitative RT-PCR has been performed. Could Western blotting be performed to have direct information on the protein levels? Figure 2C demonstrates that this is possible in cells from human patients, so why are there no data on the majority of other experiments in this manuscript?

We had already conducted Western blot analysis in the past to estimate the extent of TRIM37 depletion upon siRNA treatment (Balestra et al, 2013). However, following the suggestion of the reviewer, we repeated this analysis for select experiments of this study (HeLa Centrin-1:GFP (Figure 1—figure supplement 1A), HeLa (Figure 1—figure supplement 1B), as well as upon double depletion of TRIM37 and Centrobin (Figure 5—figure supplement 1H). This is in addition to the patient data reported already in the initial submission (currently Figure 1—figure supplement 1E).

Moreover, what is the transfection efficiency in the siRNA experiments? Is there variability between cells that might explain variability in the cenpas phenotypes?

Unfortunately, in the absence of a working antibody detecting endogenous TRIM37 by immunofluorescence analysis, we cannot provide an accurate value for the transfection efficiency in this experiment. However, given the new Western blot analysis reported in Figure 1—figure supplement 1A, B and Figure 5—figure supplement 1E, which demonstrates near complete TRIM37 depletion, the transfection efficiency must have been extremely high.

Minor point:It is stated that centrobin in si-TRIM37 cells migrates slower (Figure 4D), suggesting that TRIM37 regulates the post-translational state of centrobin. It looks to me as if the corresponding gel in Figure 4D was 'smiling' (see curvature of centrobin in the neighboring lane). I think that the authors should tone down their statement, or replace Figure 4D with a more convincing image.

We thank the reviewer for having noticed this. We now show a different gel that is not “smiling”, and in which the apparent difference in migration is also observed (Figure 4I). Moreover, we toned down the importance of this apparent difference in the text of the revised manuscript.

Reviewer #1 (Significance (Required)):The findings of this manuscript are highly significant for our understanding of centriole biogenesis. They should be of interest to a large community of cell biologists working on mitosis and on the centrosome, and they are of further importance for biomedical research related to developmental growth abnormalities (Mulibrey nanism). The manuscript shows for the first time a mechanistic link between TRIM37-dependent control of centrobin protein levels, and their impact on the formation of centriole precursors during the cell cycle. The manuscript is well presented, and the relevant scientific literature is cited correctly.However, I would prefer that a potential relationship between cenpas, tigers, and the well-described rootletin-containing striated fibres be discussed, if not controlled by additional experiments.

We thank the reviewer for her/his appreciation of our work and the support for publication.

Reviewer #2 (Evidence, reproducibility and clarity (Required)):In this work, the authors investigated roles of TRIM37 in regulation of centriole numbers. It was previously observed that depletion of TRIM37 results in supernumerary centrioles and centriole-like structures (Balestra et al., 2013; Meitinger et al., 2016). Here, the authors characterized these centriolar protein assemblies (Cenpas). Cenpas were formed, following an atypical de novo pathway and eventually trigger centriole assembly. They observed that Centrobin is frequently present in Cenpas from the early stage and other centriolar components are sequentially recruited. Furthermore, they established that Cenpas formation upon TRIM37 depletion requires PLK4 activity. TRIM37 depletion also activates PLK1-dependent centriole multiplication.1) They propose that the tiger structure acts as platform for PLK4-dependent Cenpas assembly. Cenpas may evolve into centriole-like structures after a stepwise incorporation of other centriolar proteins. Figure 6E suggests that a series of events seem to occur within G2 phase. Therefore, this reviewer suggests to perform a detailed time-course experiments at G2 phase. According to the model, the Centrobin-positive tiger structures may appear first, and then a Centrobin- and centrin-2-double positive structure starts to appear.

We thank the reviewer for suggesting this important experiment, which we have performed for the revision. Cells were treated with a double thymidine block, released from the block, and analyzed at successive time points thereafter as they progressed through S and into G2, using antibodies against Centrobin and Centrin-2. Importantly, as expected, we found that Centrobin “tiger” structures appear before Cenpas marked by Centrin-2, in line with the model. This new result is reported in the revised manuscript in Figure 5E, F, as well in the accompanying text.

2) They claim that Mulibrey patient cells exhibited evidence of chromosome mis-segregation, as would be expected from multipolar spindle assembly, and conclude that Cenpas are present and active also in Mulibrey patient cells. Chromosome mis-segregation may be observed in the normal cells, too. Therefore, they have to perform statistical analysis on Figure 2D.

In response to this suggestion and to the related comment of reviewer 3 (see below), we conducted additional immunofluorescence analysis and quantification of patient and control fibroblasts, assessing the distribution of Centrin-2, Centrobin, microtubules and *γ* -tubulin, as well as the extent of chromosome mis-segregation. In particular, to address the point raised here, we now report a quantification of the extent of chromosome mis-segregation and of cells harboring micronuclei, finding a ~20 fold elevation in each case in patient fibroblasts compared to the control. These important new findings with Mulibrey patient fibroblasts fully support the results obtained with tissue culture cells depleted of TRIM37 and are reported in the new Figure 2H-J, as well as in the accompanying text.

3) In Figure 2A, They claimed that mitotic microtubules were disrupted with the cold treatment for 30 min. In our experience, cold treatment for 30 min is not sufficient to disrupt mitotic microtubules. They may show control panel before microtubule regrowth.

The control panels have been included in the revised manuscript as requested (new panel Figure 1—figure supplement 2H).

Reviewer #2 (Significance (Required)):

*Significance of this work resides in identification and description of Cenpas as a novel centriole assembly pathway. The authors used cutting-edge microscopy techniques to visualize Cenpas. The manuscript raised more questions than answers. Nonetheless, it is worth to publish the manuscript after revision*.

We thank the reviewer for supporting publication of this work after revision.

Reviewer #3 (Evidence, reproducibility and clarity (Required)):Balestra and colleagues investigate the function of Trim 37 in centrosome biogenesis. Trim 37 is a ubiquitin ligase that has previously been identified by the authors as a regulator of centriole duplication. Mutations in Trim37 cause a rare syndrome named Mulibrey that is responsible for a severe form of dwarphism Here they show that depletion of Trim37 in human cells results in the assembly of structures that they name Cenpas. They follow the possibility that Trim37 localises to the centrosome, which might inhibit the assembly of these structures. Further they show that Trim37 depleted cells (or in patient fibroblasts ) assemble multipolar mitosis. Further analysis shows that what the authors defined as abnormal centriole structures are formed in Trim37 depleted cells. These structures recruit centrobin, a daughter centriole component and this process requires the activity of PLK4 and PLK1.Major comments:This study characterizes Trim37 and its possible role in centriole biogenesis. Most conclusions are convincing, although some of the claims taken by the authors might require more data to be corroborated.1)The major point to be taken into consideration in my opinion relates with the Cenpas structure. According to the beautiful cryo-EM data shown on Figure 3, I wonder why the authors describe these structures as centriole like- or centriole related. I think these appear very different from centrioles and this might be even quite interesting if these structures nucleate microtubules and can participate in mitotic spindle assembly.

We thank the reviewer for her/his view on this question. However, our opinion remains slightly different on this point. In our view, most of the structures we used to refer as “centriole-like” or “centriole-related” indeed resemble the centriole organelle at the ultrastructural level, notably in containing microtubule bundles and in being of a related size (in addition to bearing centriolar markers). This being written, we recognize that the distinction between the two types of structures was somewhat arbitrary, and have combined them into the more prudent term “centriole-related” in the revised manuscript. Moreover, we further explain that these entities comprise a range of ultrastructures.

The authors correlate these non-canonical centriole structures as possible microtubule nucleators that might be responsible for multipolar configurations like in Figure 2D. This correlation has to be established. In Figure 2D, the authors analyze configurations of mitotic cells in terms of centrosome number and characterized frequency of extra foci. To me the foci they show are quite different in nature. Poles 1 and 3 have both centrin and γ-tubulin (presumably centrioles), pole 2 has only a tiny amount of centrin and no γ-tubulin, while pole 4 appears to contain both but less of each protein. So the question is are they all nucleating microtubules and participating in spindle assembly? This is particularly important in light of what the authors then mention, which is the occurrence of chromosome mis-segreation in patient cells (this is not shown). Also they describe these extra poles, and then say that Cenpas are active in patient cells. But, active in which manner? By nucleating microtubules? First, in either siRNA cells or in patient cells the authors should analyze microtubules and show that all the extra poles (made of non-canonical centriole) nucleate microtubules and participate in spindle assembly.

In response to this suggestion and to the related comment of reviewer 2 (see above), we conducted additional immunofluorescence analysis and quantification of patient and control fibroblasts, assessing the distribution of Centrin-2, Centrobin, microtubules and *γ*-tubulin, as well as the extent of chromosome mis-segregation. Although there are indeed differences between individual spindle poles, these experiments demonstrate that Cenpas in patient fibroblasts can nucleate microtubules, resulting in multipolar spindle assembly and chromosome mis-segregation. Moreover, we found Centrobin “tiger” structures also in patient cells. These important new findings with Mulibrey patient fibroblasts fully support the results obtained with tissue culture cells deplete of TRIM37 and are reported in the new Figure 2C-J and Figure 5A-D, as well as in the accompanying text.

If they want to propose that this might be the cause of genome integrity loss in patients (as stated in the Abstract and suggested a few times throughout the paper) they have to show that cells divide abnormally and generate aneuploidy progeny.

See response just above.

2) Another important point that is only partially addresses is the function of Trim37 in stabilizing centrobin. Does Trim37 ubiquitinates centrobin? While the western blot on Figure 4 shows an increase at 8hrs in Trim37 RNAi, this is also the case for tubulin (Figure 4E). But the overall levels appear only slightly increased when compared to its levels at time point zero (Figure 4F). I can see that in siRNA Ctrl Trim 37 levels go down, but it is still present so how do they explain the lack of Cenpas in this case? Is there a threshold that supports centriole duplication without any major defect but accumulation of a certain level of centrobin then generates Cenpas? Can the authors generate Cenpas just by over-expressing centrobin directly?

It appears from the comment of the reviewer that our explanations were not sufficiently clear. The experiment reported in the original Figure 4E and 4F (currently Figure 4J and 4K) was done in the presence of cycloheximide to analyze the half-life of Centrobin in control conditions and upon TRIM37 depletion. We have clarified the text in the revised manuscript to facilitate understanding of this point. Moreover, we have repeated this experiment to solidify this finding.

As for the question about the generation of Cenpas upon Centrobin overexpression: this experiment was already reported in the initial submission (now Figure 4—figure supplement 2B), and shows that such overexpression is not sufficient to generate Cenpas, so that it is not merely the level of Centrobin that matters for their production.

So in conclusion, the link between Cenpas and multipolarity has to be better investigated in my opinion. This should not be time consuming and also not extremely costly. Authors should label spindle MTs in patient fibroblasts to show that indeed Cenpas are nucleating microtubules. Ideally Cenpas would be distinguished by centrobin labeling. In siRNA depleted cells maybe time lapse microscopy can be used to image mitosis and show a correlation between Cenpas and multipolarity?

See response above to point 1). The suggestion to conduct time-lapse imaging to further clarify the relationship between Cenpas and multipolarity does not seem needed in light of the new findings with patient fibroblasts, which showed that both Cenpas and Centrobin structures frequently function as supernumerary MTOCs during mitosis (Figure 2C, D and Figure 5C, D).

The data is presented without statistical analysis on the figures only on figure legends, This is really difficult for the reader. The number of experiments and cells analyzed maybe should be also included in each figure.

We had kept this information in the legends in the initial submission merely to have lean figures. Prompted by the reviewer’s comment, we now turned to a letter system throughout the figures to systematically report in a graphically succinct and comprehensible manner the significance of all comparisons within each experiment.

Minor comments:Some picture lack scale bars

Apologies -this has been fixed.

the localization of GFP-Trim37. On Figure 1 the authors describe a different localization when fused to a NES localization. It is true that a dotty signal is seen on the panel of NES (Figure 1D), but a nuclear signal is not seen on Trim-GFP in any of the images provided. Shouldn't this be the case?

In fact, there is indeed some nuclear TRIM37:GFP signal in the left panel of Figure 1D, but it is very weak. To make this more apparent, we now provide insets for the two cells with matched and adjusted brightness/contrast to emphasize this point (Figure 1—figure supplement 2A).

Figure 1C is missing a siCtrl.

Thank you for spotting this; the control quantification has been added to Figure 1C.

Why Trim37GFP does not rescue completely the assembly of the extra foci?

There can be many reasons why rescue would not be complete in such an experimental setting, be it for TRIM37 or another protein, including slightly different protein levels, distribution, or interaction with partner proteins.

In Figure 6E, are the authors sure that in the condition of siTRim3 plus si Centrobin and Plk1 inhibition, cells are not stuck in S-phase? This might explain the lack of being in a permissive G2 phase to generate Cenpas?

Although Plk1 inhibition was not expected to block cells in S phase based on the extant literature, we could not rule out this possibility from the data in the initial submission. Therefore, prompted by the comment of the reviewer, we conducted FACS analysis following release from a double thymidine block of cells depleted of TRIM37, Centrobin, TRIM37+Centrobin, as well as of cells depleted of TRIM37 or TRIM37+Centrobin subjected to BI2536. As reported in the new Figure 6—figure supplement 1B, PLK1 inhibition by BI2536 indeed does not block cells in S phase.

The data is presented without statistical analysis on the figures. This can be found on figure legends, but it is better to include on the figures to facilitate the reader's job. The number of experiments and cells analyzed maybe should be also included in each figure?

See point above.

Reviewer #3 (Significance (Required)):Interesting findings and quite novel since a role for Trim 37 in centriole biogenesis has never been reported. Also quite interesting the possible link between multipolarity (needs better characterization) and Mulibrey syndrome.

We thank the reviewer for recognizing the interest and novelty of our work.